# Distributionally Robust Performative Prediction

**Songkai Xue**
Department of Statistics
University of Michigan
sxue@umich.edu

**Yuekai Sun**
Department of Statistics
University of Michigan
yuekai@umich.edu

## Abstract

Performative prediction aims to model scenarios where predictive outcomes subsequently influence the very systems they target. The pursuit of a performative optimum (PO)—minimizing performative risk—is generally reliant on modeling of the *distribution map*, which characterizes how a deployed ML model alters the data distribution. Unfortunately, inevitable misspecification of the distribution map can lead to a poor approximation of the true PO. To address this issue, we introduce a novel framework of distributionally robust performative prediction and study a new solution concept termed as distributionally robust performative optimum (DRPO). We show provable guarantees for DRPO as a robust approximation to the true PO when the nominal distribution map is different from the actual one. Moreover, distributionally robust performative prediction can be reformulated as an augmented performative prediction problem, enabling efficient optimization. The experimental results demonstrate that DRPO offers potential advantages over traditional PO approach when the distribution map is misspecified at either micro- or macro-level.

## 1 Introduction

In numerous fields where predictive analytics play an important role, decisions made on the basis of machine learning models do not just passively predict outcomes but actively influence future input data. Consider the domain of financial services, such as credit scoring and loan issuance, where a model's decision to grant or deny an application can affect the applicant's future financial behaviors and, consequently, the profile of future applicants. Similarly, in educational settings, the decision process for school admissions can shape the applicant pool, as those who are accepted often share their success strategies, indirectly influencing the preparation of future candidates. These examples highlight the study of performative prediction [29], a recent framework that facilitates a formal examination of learning in the presence of performative distribution shift resulting from deployed ML models.

Delving deeper into the formulations of performative prediction, the concept of a distribution map emerges as pivotal. This map characterizes the impact that a deployed ML model has on the underlying data distribution, which is a crucial element in navigating performative effects. The literature primarily revolves around the pursuit of performative stability (PS), a model which is optimal for the distribution it induces [29, 25, 5, 8]. However, a more ambitious target is the performative optimum (PO), which seeks the minimization of performative risk, the risk of the deployed model on the distribution it induces. Efficiently achieving PO typically necessitates a modeling of the distribution map [26, 14, 15, 24]. Practically, the precise influence of a model on the data ecosystem is intricate and dynamic, making perfect specification an unattainable ideal.

In this work, we propose a distributionally robust performative prediction framework that aims to enhance robustness against a spectrum of distribution maps, thereby mitigating the issue of misspecification. Our contributions are summarized as follows: 1) in Section 2, we formalize the

38th Conference on Neural Information Processing Systems (NeurIPS 2024).

DRPO concept, anchoring it within the performative prediction literature as a robust alternative; 2) in Section 3, we provide theoretical insights into the efficacy of DRPO, demonstrating its resilience in the face of distribution map misspecification; 3) in Section 4, we recast distributionally robust performative risk minimization as an augmented performative risk minimization problem, facilitating efficient optimization; 4) in Section 5, we showcase DRPO's advatanges over conventional PO by empirical experiments. The paper concludes with a summary and discussion.

## 1.1 Related Work

**Performative prediction.** Performative prediction is an emerging framework for learning models that influence the data they intend to predict. The majority of research focuses on performative stability [29, 25, 5, 30, 21, 8, 7], albeit with a few exceptions aiming at performative optimality. [26] propose a two-stage plug-in method for finding the PO. [14] propose to find the PO by a parametric model of the distribution map. This method is extended by [15] to stateful performative prediction. [18] solves the PO when the problem is outcome performative only. [24] argue that the PO with a misspecified nominal distribution map can still reasonably approximate the true PO, as long as the misspecification level is not significant. This claim is also supported by our theory and experimental findings. Unlike them, we demonstrate that the DRPO is comparable to the PO if the misspecification is small, whereas the DRPO can offer substantial advantages over the PO if the misspecification is moderate to large. Moreover, the DRPO ensures reasonable performance for all distribution maps in an uncertainty collection surrounding the nominal distribution map, rather than just a single distribution map.

**Distributionally robust optimization.** DRO solves a stochastic optimization problem by optimizing under the worst-case scenario over an uncertainty set of probability distributions. Most popular DRO frameworks are based on $\varphi$-divergence [13, 2, 11, 12, 20, 9] and Wasserstein distance [11, 27, 6, 31, 4, 3, 10]. The existing DRO literature pays no attention to performative prediction except for [28]. [28] study the repeated distributionally robust optimization algorithm, which repeatedly minimizes the distributionally robust risk at the induced distribution. They show such repeated training algorithm yields a distributionally robust performative stable (DRPS) solution, assuming conditions analogous to the validity of repeated risk minimization [29, 25] in finding the performative stable (PS) solution. Nevertheless, the DRPO solution, which lies at the heart of the distributionally robust performative prediction problem, is not the subject of their study. Furthermore, they lacks theoretical guarantees regarding the proximity of the DRPS to either the PS or PO solution.

## 2 Methodology

### 2.1 Performative Prediction Essentials

Let $\Theta$ denote the (finite-dimensional) model parameter space, $\mathcal{Z}$ denote the data sample space, and $\mathcal{P}(\mathcal{Z})$ denote the set of probability measures supported on $\mathcal{Z}$. In performative prediction, we aim to find a $\theta \in \Theta$ that achieves low *performative risk*

$$\mathrm{PR}_{\mathrm{true}}(\theta) = \mathbb{E}_{Z \sim \mathcal{D}_{\mathrm{true}}(\theta)}[\ell(Z; \theta)], \tag{2.1}$$

where $\ell : \mathcal{Z} \times \Theta \to \mathbb{R}$ is the (known) loss function, and $\mathcal{D}_{\mathrm{true}} : \Theta \to \mathcal{P}(\mathcal{Z})$ is the *true distribution map*. The *true performative optimum* (true PO) $\theta_{\mathrm{PO,true}}$ is known to minimize the (true) *performative risk*:

$$\theta_{\mathrm{PO,true}} \in \arg\min_{\theta \in \Theta} \left\{ \mathrm{PR}_{\mathrm{true}}(\theta) = \mathbb{E}_{Z \sim \mathcal{D}_{\mathrm{true}}(\theta)}[\ell(Z; \theta)] \right\}. \tag{2.2}$$

It is easy to see that if we *know* the true distribution map, then we can evaluate the true performative risk and find the true performative optimum well up to some finite sample error which is negligible as the sample size goes to infinity. However, the true map $\mathcal{D}_{\mathrm{true}}(\cdot)$ is *unknown* in general, thus posing a significant obstacle in the pursuit of evaluating and optimizing the true performative risk.

To enable the optimization of performative risk, it is necessary to have a known *nominal distribution map* $\mathcal{D}(\cdot)$ that is believed to closely approximate the unknown true distribution map $\mathcal{D}_{\mathrm{true}}(\cdot)$. Then one can find the *performative optimum* (PO) by minimizing the *nominal performative risk*:

$$\theta_{\mathrm{PO}} \in \arg\min_{\theta \in \Theta} \left\{ \mathrm{PR}(\theta) = \mathbb{E}_{Z \sim \mathcal{D}(\theta)}[\ell(Z; \theta)] \right\}. \tag{2.3}$$

Because the distribution map is inevitably *misspecified*, $\mathcal{D}(\cdot) \neq \mathcal{D}_{\mathrm{true}}(\cdot)$, the true performative risk is generally not minimized by $\theta_{\mathrm{PO}}$. Therefore, we treat (2.3) as a solution concept to approximately

solve the true performative risk minimization problem (2.2), and refer (2.3) to *standard performative prediction*. Now we provide several illustrative instances of potential sources that may lead to the misspecification of distribution map, *i.e.*, $\mathcal{D}(\cdot) \neq \mathcal{D}_{\text{true}}(\cdot)$.

**Modeling error.** The modeling of $\mathcal{D}(\cdot)$ can be either a *deterministic model* of explicit form or a *statistical model* with model parameters to be estimated. The misspecification of $\mathcal{D}(\cdot)$ may stem from modeling error.

**Example 2.1** (Strategic classification). *Let $\mathcal{Z} = \mathcal{X} \times \mathcal{Y}$ where $\mathcal{X}$ is the feature space and $\mathcal{Y}$ is the label space, so that we are in the supervised learning regime. Strategic classification relies on a working model of individual's data manipulation strategy:*

$$\Delta_\theta(x) = \arg\max_{x'}\{u_\theta(x') - c(x, x')\}, \tag{2.4}$$

*where $\Delta_\theta(\cdot)$, $u_\theta(\cdot)$, and $c(\cdot, \cdot)$ are known as individual's best response function, utility function, and cost function, respectively. The best response function can be lifted to the measurable space of $\mathcal{X} \times \mathcal{Y}$ so that we have the response map*

$$T_\theta\left(\begin{bmatrix} x \\ y \end{bmatrix}\right) = \begin{bmatrix} \Delta_\theta(x) \\ y \end{bmatrix} = \begin{bmatrix} \arg\max_{x'}\{u_\theta(x') - c(x, x')\} \\ y \end{bmatrix}.$$

*The nominal distribution map $\mathcal{D}(\cdot)$ is fully characterized by $T_\theta(\cdot)$ and the sampling distribution of $\mathcal{D}_{\text{true}}(\mathbf{0})$ (see details in Appendix E). In this example, the distribution map $\mathcal{D}(\cdot)$ could be misspecified because the individual's utility function and cost function could be misspecified.*

**Example 2.2** (Location family). *Location family postulates a translation model:*

$$Z \sim \mathcal{D}(\theta) \iff Z \overset{d}{=} Z_0 + A\theta \text{ where } Z_0 \sim \mathcal{D}_{\text{true}}(\mathbf{0}),$$

*where $A \in \mathbb{R}^{\dim(\mathcal{Z}) \times \dim(\Theta)}$ is unknown and therefore must be estimated. If we observe the sampling distributions of $\mathcal{D}(\theta_0), \mathcal{D}(\theta_1) \dots, \mathcal{D}(\theta_K)$, then $A$ is partially identified up to a linear subspace of $\mathbb{R}^{\dim(\mathcal{Z}) \times \dim(\Theta)}$:*

$$A \in \{M \mid M(\theta_k - \theta_0) = \mu_k - \mu_0 \text{ for } k \in [K]\},$$

*where $\mu_k$ is the mean of $\mathcal{D}(\theta_k)$. In this example, the distribution map $\mathcal{D}(\cdot)$ can be misspecified because the model parameter $A$ is only partially identifiable.*

**Distribution shift.** Consider the training and test environments have different distribution maps, $\mathcal{D}_{\text{train}}(\cdot)$ and $\mathcal{D}_{\text{test}}(\cdot)$, respectively. We specify the nominal distribution map as the training distribution map $\mathcal{D}(\cdot) = \mathcal{D}_{\text{train}}(\cdot)$. Then $\mathcal{D}(\cdot)$ can be misspecified for the true distribution map $\mathcal{D}_{\text{true}}(\cdot) = \mathcal{D}_{\text{test}}(\cdot)$ due to the difference between the training and test environments $\mathcal{D}_{\text{train}}(\cdot) \neq \mathcal{D}_{\text{test}}(\cdot)$.

**Example 2.3** (Disparate impacts and fairness). *A population is comprised of majority and minority subpopulations (e.g., by race or gender). The population distribution map is a mixture of the subpopulation distribution maps: $\mathcal{D}_{\text{pop}}(\theta) \overset{d}{=} \gamma \mathcal{D}_{\text{maj}}(\theta) + (1 - \gamma)\mathcal{D}_{\text{min}}(\theta)$. In fair machine learning, a theme is to check whether an ML model has disparate impacts on different subpopulations or biased against the minority. Suppose that we work with $\mathcal{D}(\cdot) = \mathcal{D}_{\text{pop}}(\cdot)$ and target at the minority $\mathcal{D}_{\text{true}}(\cdot) = \mathcal{D}_{\text{min}}(\cdot)$. However, the population distribution map and the minority distribution map may differ. In this example, the distribution map $\mathcal{D}(\cdot)$ is misspecified because of subpopulation shift.*

## 2.2 Distributionally Robust Performative Prediction

From an intuitive perspective, the PO solution (2.3) has the potential to achieve low performative risk $\text{PR}_{\text{true}}(\theta_{\text{PO}})$ when the nominal distribution map $\mathcal{D}(\cdot)$ closely aligns with the true distribution map $\mathcal{D}_{\text{true}}(\cdot)$. However, $\mathcal{D}(\cdot)$ and $\mathcal{D}_{\text{true}}(\cdot)$ may quite different. In such cases, the PO solution may incur high performative risk. To address this issue, we propose a distributionally robust formulation for performative prediction, where we explicitly incorporate into the learning phase the consideration that the true distribution map $\mathcal{D}_{\text{true}}(\cdot)$ is different from the nominal one $\mathcal{D}(\cdot)$.

Let $D(\cdot \| \cdot)$ be the KL divergence, *i.e.*,

$$D(Q\|P) = \int \varphi\left(\frac{dQ}{dP}\right) dP = \int \log\left(\frac{dQ}{dP}\right) dQ,$$

where $\varphi(t) = t \log t$ for any $t > 0$. Here $dQ/dP$ is the Radon–Nikodym derivative, and we implicitly require the probability measure $Q$ to be absolutely continuous with respect to $P$. With the

KL divergence, we can define a family of distribution maps around the nominal distribution map. Specifically, the *uncertainty collection* around $\mathcal{D}$ with *critical radius* $\rho$ is defined as

$$\mathcal{U}(\mathcal{D}) = \{\widetilde{\mathcal{D}} : \Theta \to \mathcal{P}(\mathcal{Z}) \mid D(\widetilde{\mathcal{D}}(\theta)\|\mathcal{D}(\theta)) \leq \rho, \forall \theta \in \Theta\}.$$

The radius $\rho$ reflects the the magnitude of shifts in distribution map we seek to be robust to. We remark that the value of $\rho$ can be prescribed or be selected in data-driven ways. We postpone the discussion on critical radius calibration to Section 4.3.

**Definition 2.4** (Distributionally robust performative risk)**.** *The distributionally robust performative risk with the uncertainty collection $\mathcal{U}(\mathcal{D})$ is defined as*

$$\mathrm{DRPR}(\theta) = \sup_{\widetilde{\mathcal{D}} : \widetilde{\mathcal{D}} \in \mathcal{U}(\mathcal{D})} \mathbb{E}_{Z \sim \widetilde{\mathcal{D}}(\theta)}[\ell(Z; \theta)]. \tag{2.5}$$

In other words, the distributionally robust performative risk $\mathrm{DRPR}(\theta)$ measures the worst possible performative risk incurred by the model parameterized by $\theta$ among the collection of all alternative distribution maps that are $\rho$-close to the nominal distribution map $\mathcal{D}$. With this intuition, it is natural to define an alternative solution concept which minimizes (2.5).

**Definition 2.5** (Distributionally robust performative optimum)**.** *The distributionally robust performative optimum (DRPO) is defined as*

$$\theta_{\mathrm{DRPO}} \in \arg\min_{\theta \in \Theta} \mathrm{DRPR}(\theta). \tag{2.6}$$

We refer the method (2.6) to *distributionally robust performative prediction*. When comparing (2.6) and (2.3), we view the DRPO (2.6) as a competing solution concept to the PO (2.3), because both of them aim for achieving low performance risk (2.1).

## 2.3 Generalization Principle of DRPO

Distributionally robust performative prediction asks to not only perform well on a fixed performative prediction problem (parameterized by the distribution map $\mathcal{D}$), but simultaneously for a range of performative prediction problems, each determined by a distribution map in an uncertainty collection $\mathcal{U}$. This results in more robust solutions, that is, those DRPOs which are robust to misspecification of distribution map. The uncertainty collection plays a key role: it implicitly defines the induced notion of robustness. Moreover, distributionally robust performative prediction yields a natural approach for certifying out-of-sample performance, which is summarized by the following principle.

**Proposition 2.6** (Generalization principle of distributionally robust performative prediction)**.** *Suppose that the uncertainty collection $\mathcal{U}$ contains the true distribution map $\mathcal{D}_{\mathrm{true}}$, then the true performative risk is bounded by the distribution robust performative risk:* $\mathrm{PR}_{\mathrm{true}}(\theta) \leq \mathrm{DRPR}(\theta)$ *for any $\theta \in \Theta$. In consequence, we have* $\mathrm{PR}_{\mathrm{true}}(\theta_{\mathrm{DRPO}}) \leq \mathrm{DRPR}(\theta_{\mathrm{DRPO}})$.

Essentially, if $\mathcal{U}$ is chosen appropriately, the corresponding DR performative risk upper bounds the true performative risk, and thus DRPO enjoys provable guarantees on its incurred performative risk.

## 2.4 Benefits of DRPO: A Toy Example

A toy example is employed to facilitate a conceptual understanding of the advantages of DRPO over PO in relation to achieving improved control over the worst-case performative risk.

Let $\mathcal{Z} = \mathbb{R}$, $\Theta = [-1, 1]$, and $\ell(z; \theta) = \theta z$. Let the nominal distribution map be $\mathcal{D}(\theta) = \mathcal{N}(f(\theta), \sigma^2)$ for some $f : [-1, 1] \to \mathbb{R}$ and $\sigma^2 > 0$. Then the nominal performative risk is

$$\mathrm{PR}(\theta) = \mathbb{E}_{Z \sim \mathcal{N}(f(\theta), \sigma^2)}[\ell(Z; \theta)] = \theta f(\theta).$$

**Regularization effect.** With the dual formula given in Section 3.1, one can derive the distributionally robust performative risk directly:

$$\mathrm{DRPR}(\theta) = \mathrm{PR}(\theta) + \sqrt{\rho}\,\mathrm{Penalty}(\theta),$$

where the penalty function $\mathrm{Penalty}(\theta) = \sqrt{2\sigma^2}\,|\theta|$ penalizes the deviation of $\theta$ from the origin 0, and the critical radius $\rho$ tunes the level of regularization. That is to say, in this toy example, the distributionally robust performative risk minimization problem is essentially a $L_1$-*regularized* performative risk minimization problem.

**Better worst-case control.** To be more concrete, we let $f(\theta) = a_1\theta + a_0$ for some $a_1, a_0 > 0$. For any $\widetilde{\mathcal{D}} \in \mathcal{U}(\mathcal{D})$, let the performative risk of $\widetilde{\mathcal{D}}$ be $\mathrm{PR}_{\widetilde{\mathcal{D}}}(\theta) = \mathbb{E}_{Z \sim \widetilde{\mathcal{D}}(\theta)}[\ell(Z;\theta)]$. If $\widetilde{\mathcal{D}}$ is the true distribution map, then $\mathrm{PR}_{\widetilde{\mathcal{D}}}(\theta)$ is the true performative risk that we incur. Through direct calculation (see details in Appendix A), one can show that $\theta_{\mathrm{DRPO}}$ is more robust than $\theta_{\mathrm{PO}}$ in the sense of *worst-case performative risk* control, that is, $\sup_{\widetilde{\mathcal{D}} \in \mathcal{U}(\mathcal{D})} \mathrm{PR}_{\widetilde{\mathcal{D}}}(\theta_{\mathrm{PO}}) \geq \sup_{\widetilde{\mathcal{D}} \in \mathcal{U}(\mathcal{D})} \mathrm{PR}_{\widetilde{\mathcal{D}}}(\theta_{\mathrm{DRPO}}) + \frac{\rho\sigma^2}{2a_1}$ for any fixed $\rho \leq \frac{a_0^2}{2\sigma^2}$. This example shows that the worst-case performative risk of the PO can be *arbitrarily larger than* that of the DRPO, as $a_1 \to 0$. The message that DRPO offers certain advantages over PO in terms of mitigating worst-case performative risk is also supported by the empirical results shown in Section 5.

# 3 Theory

## 3.1 Strong Duality of DRPR

The evaluation of distributionally robust performative risk $\mathrm{DRPR}(\theta)$ given in (2.5) involves an infinite dimensional maximization problem which is generally intractable. Fortunately, it is in fact equivalent to a minimization problem over a single dual variable.

**Proposition 3.1** (Strong duality of DRPR). *For any $\theta \in \Theta$, we have*

$$\mathrm{DRPR}(\theta) = \inf_{\mu \geq 0} \left\{ \mu \log \mathbb{E}_{Z \sim \mathcal{D}(\theta)}\left[ e^{\ell(Z;\theta)/\mu} \right] + \mu\rho \right\}. \tag{3.1}$$

The dual reformulation (3.1) will be served as the cornerstone of developing algorithms for finding the DRPO in Section 4. As a byproduct of Proposition 3.1, a characterization of the worst-case distribution map which attains the supremum in (2.5) is given in Appendix B.

## 3.2 Excess Risk Bound

For now, we are interested in bounding the excess risk: $\mathcal{E}(\hat{\theta}) = \mathrm{PR}_{\mathrm{true}}(\hat{\theta}) - \min_{\theta \in \Theta} \mathrm{PR}_{\mathrm{true}}(\theta) = \mathrm{PR}_{\mathrm{true}}(\hat{\theta}) - \mathrm{PR}_{\mathrm{true}}(\theta_{\mathrm{PO,true}})$, where $\hat{\theta}$ is an approximate solution to the true PO $\theta_{\mathrm{PO,true}}$, which could particularly be $\theta_{\mathrm{PO}}$ and $\theta_{\mathrm{DRPO}}$. The excess risk captures the true performance of $\hat{\theta}$ relative to the oracle performative optimum $\theta_{\mathrm{PO,true}}$, providing a direct measurement of the suboptimality of $\hat{\theta}$ in terms of performative risk. As follows, we show the excess risk bounds of the PO solution and the DRPO solution, $\mathcal{E}(\theta_{\mathrm{PO}})$ and $\mathcal{E}(\theta_{\mathrm{DRPO}})$, and compare them.

**Proposition 3.2** (Excess risk bound of the PO). *Suppose that we have bounded loss function $|\ell(z;\theta)| \leq B$ for any $z \in \mathcal{Z}, \theta \in \Theta$ and some $B > 0$. Then we have*

$$\mathcal{E}(\theta_{\mathrm{PO}}) \leq \sqrt{2}B \sup_{\theta \in \Theta} \sqrt{D(\mathcal{D}_{\mathrm{true}}(\theta)\|\mathcal{D}(\theta))}. \tag{3.2}$$

**Proposition 3.3** (Excess risk bound of the DRPO). *Suppose that $D(\mathcal{D}_{\mathrm{true}}(\theta_{\mathrm{DRPO}})\|\mathcal{D}(\theta_{\mathrm{DRPO}})) \leq \rho$. Then we have*

$$\mathcal{E}(\theta_{\mathrm{DRPO}}) \leq \sqrt{\rho \mathrm{Var}_{Z \sim \mathcal{D}(\theta_{\mathrm{PO,true}})}[\ell(Z;\theta_{\mathrm{PO,true}})]} + o(\sqrt{\rho}). \tag{3.3}$$

Comparing Proposition 3.2 and 3.3, we see that the excess risk bound of the DRPO can be localized to the true PO while the excess risk bound of the PO is entangled with the full parametric space $\Theta$. Although we are comparing two upper bounds which can be not tight enough, the comparison sheds lights to the potential benefits of using DRPO over PO solution. Even if in the case of no significant improvement of using DRPO over PO solution, the excess risks of them are comparable, thus doing no harm. Our insight has been verified through a toy example in Section 2.4 and as well experimental results in Section 5. In Appendix C, we generalize Proposition 3.3 to the scenario when the uncertainty collection doesn't cover the true distribution map, *i.e.*, $D(\mathcal{D}_{\mathrm{true}}(\theta_{\mathrm{DRPO}})\|\mathcal{D}(\theta_{\mathrm{DRPO}})) > \rho$.

# 4 Algorithms

We recall a standard algorithm for performative risk minimization in Appendix E. In the following subsections, we will see that any off-the-shelf algorithms for finding the PO can be utilized as an *intermediate algorithm* for finding the DRPO by using our proposed algorithms. Moreover, we provide practical considerations for the selection of a critical radius in the last subsection.

## 4.1 Distributionally Robust Performative Risk Minimization

By the strong duality in Proposition 3.1, DR performative risk minimization is equivalent to the following optimization problem jointly over $(\theta, \mu)$:

$$\min_{\theta} \mathrm{DRPR}(\theta) \iff \min_{\theta \in \Theta} \inf_{\mu \geq 0} \left\{ \psi(\theta, \mu) = \mu \log \mathbb{E}_{Z \sim \mathcal{D}(\theta)} \left[ e^{\ell(Z;\theta)/\mu} \right] + \mu \rho \right\}. \tag{4.1}$$

This suggests an alternating minimization, summarized in Algorithm 1, where we learn $\theta_{\mathrm{DRPO}}$ by fixing $\mu$ and minimizing on $\theta$ and then fixing $\theta$ and minimizing on $\mu$ alternatively until convergence.

The step of minimizing on $\theta$ with fixed $\mu$ (Line 4 in Algorithm 1) is itself a performative risk minimization problem, which can be solved by any suitable performative risk minimization algorithm. The step of minimizing on $\mu$ with fixed $\theta$ (Line 5 in Algorithm 1) can be solved by the line search or the Newton–Raphson method since $\psi(\theta, \mu)$ is convex in $\mu$. The total cost of Algorithm 1 is therefore com-

---

**Algorithm 1** DR Performative Risk Minimization

1: **Input:** radius $\rho$, nominal distribution map $\mathcal{D}(\theta)$
2: Initialize $\mu$
3: **while** $\mu$ has not converged **do**
4:     Update $\theta \leftarrow \arg\min_{\theta \in \Theta} \left\{ \mathbb{E}_{Z \sim \mathcal{D}(\theta)} \left[ e^{\ell(Z;\theta)/\mu} \right] \right\}$
5:     Update $\mu \leftarrow \arg\min_{\mu \geq 0} \left\{ \psi(\theta, \mu) \right\}$ ($\psi$ in (4.1))
6: **end while**
7: **Return:** $\theta$

---

parable to that of the performative risk minimization algorithm used in the intermediate step. Lastly, the alternating minimization algorithm in common practice guarantees global convergence (to stationary point) regardless of how the optimization parameters are initialized. With the strong convexity assumption, the alternating minimization algorithm guarantees convergence to the global minimum.

## 4.2 Tilted Performative Risk Minimization

Treating $\rho$ as a hyperparameter which can be tuned by a practitioner, the solution of the dual problem (4.1) can be denoted by $(\theta^\star(\rho), \mu^\star(\rho))$. One can show that $\mu^\star(\rho)$ is a decreasing function of $\rho$. An intuition is that as $\mu \to \infty$, we have $\arg\min_{\theta \in \Theta} \{\psi(\theta, \mu)\} \approx \arg\min_{\theta \in \Theta} \{\mathbb{E}_{Z \sim \mathcal{D}(\theta)}[\ell(Z;\theta)]\}$, which reduces to the original performative risk minimization problem, or the distributionally robust performative risk minimization problem with $\rho = 0$ (see an formal explanation in Appendix F).

With this motivation, instead of tuning $\rho$, we can tune $\mu$ (or the inverse of it, denoted by $\alpha = \mu^{-1}$) and solve the $\alpha$-*tilted performative risk minimization* problem:

$$\theta_{\mathrm{TPO}} \in \arg\min_{\theta \in \Theta} \left\{ \mathrm{TPR}(\theta) = \mathbb{E}_{Z \sim \mathcal{D}(\theta)} \left[ e^{\alpha \ell(Z;\theta)} \right] \right\}, \tag{4.2}$$

where $\mathrm{TPR}(\theta)$ stands for the tilted performative risk and $\theta_{\mathrm{TPO}}$ is the *tilted performative optimum*, that is, the performative optimum of the tilted problem. In order to have stronger distributional robustness property,

---

**Algorithm 2** Tilted Performative Risk Minimization

1: **Input:** tilt $\alpha$, nominal distribution map $\mathcal{D}(\theta)$
2: Update $\theta \leftarrow \arg\min_{\theta \in \Theta} \left\{ \mathbb{E}_{Z \sim \mathcal{D}(\theta)} \left[ e^{\alpha \ell(Z;\theta)} \right] \right\}$
3: **Return:** $\theta$

---

we tune $\alpha$ to be larger. Given the correspondence $\mu^\star(\rho)$, we should have $\theta_{\mathrm{TPO}}$ with $\alpha$ equals $\theta_{\mathrm{DRPO}}$ with $\rho = (\mu^\star)^{-1}(1/\alpha)$. Therefore, the tilted performative risk minimization *implicitly* solves a corresponding DR performative risk minimization problem. Finally, we remark that exponential tilting is a statistical method that has been around at least since the exponential family [17] was first invented. More recently, it has been applied to operation research [1] and machine learning [22, 23].

## 4.3 Calibration of Critical Radius

The performance of the DRPO is contingent on the uncertainty collection radius $\rho$, which is typically difficult to choose a priori without additional information. The greater the value of $\rho$, the higher the level of distributional robustness, and thus the greater the tolerance for distribution map misspecification. Therefore, the selection of $\rho$ reflects a practitioner's risk-aversion preference. In this subsection, we present two simple, yet effective calibration techniques for selecting $\rho$.

**Post-fitting calibration.** Without additional information, we can only hope to be robust to a prescribed set of distribution maps, say $\Xi$. The assumption of Proposition 3.3 requires only the uncertainty collection at the DRPO, which reduces to an uncertainty ball centered at $\mathcal{D}(\theta_{\mathrm{DRPO}})$, encompassing the true distribution $\mathcal{D}_{\mathrm{true}}(\theta_{\mathrm{DRPO}})$. Therefore, in order to provide a provable guarantee for all distribution maps in $\Xi$, the radius $\rho$ can be chosen based on the following criterion:

$$\rho_{\mathrm{cal}} = \arg\min_{\rho \geq 0} \left\{ \max_{\mathcal{D}_{\mathrm{true}} \in \Xi} \widehat{D}(\mathcal{D}_{\mathrm{true}} \| \mathcal{D}) \leq \rho \right\}, \text{where}$$

$$\widehat{D}(\mathcal{D}_{\mathrm{true}} \| \mathcal{D}) := \widehat{D}(\mathcal{D}_{\mathrm{true}}(\theta_{\mathrm{DRPO}}(\rho)) \| \mathcal{D}(\theta_{\mathrm{DRPO}}(\rho))).$$

Here $\widehat{D}$ is the estimated KL divergence, and $\theta_{\mathrm{DRPO}}(\rho)$ is indexed by the radius $\rho$ to highlight its dependence on $\rho$ as a tuning parameter to be calibrated. We implement the post-fitting calibration approach (with bisection search for $\rho$) in Section 5.1.

**Calibration set.** With additional information, such as a small set of calibration data, we can pick $\rho$ (or $\alpha$ if we use Algorithm 2) by evaluating the performance of $\theta_{\mathrm{DRPO}}(\rho)$ on the calibration set. Consider Example 2.3 where the training and test distribution maps may differ, we can conduct the following grid searching procedure to choose $\rho$: 1) for a candidate set $\mathcal{C}$ of $\rho$'s, we compute $\{\theta_{\mathrm{DRPO}}(\rho) : \rho \in \mathcal{C}\}$ under $\mathcal{D}_{\mathrm{train}}$; 2) we obtain a few calibrating samples from $\mathcal{D}_{\mathrm{test}}$, on which we evaluate the performance of $\theta_{\mathrm{DRPO}}(\rho)$; 3) we select $\rho_{\mathrm{cal}} \in \mathcal{C}$ with the best calibration set performance. The calibration set approach is implemented in Section 5.3.

To conclude this subsection, we discuss the computational costs of the proposed calibration methods. For general problems, these calibration methods necessitate a grid search, which may be computationally expensive. Fortunately, for specific problems (for example, experiments in Section 5.1), we can exploit the decreasing nature of the estimated KL divergence as a function in $\rho$. As a result, we can use bisection search rather than grid search to significantly reduce computational costs.

# 5 Experiments

We revisit Examples 2.1, 2.2, and 2.3 and compare the PO and the DRPO empirically. For a particular (true) distribution map, either the PO or the DRPO may have the potential to outperform the other in terms of the performative risk evaluated at this particular distribution map. In contrast to the PO, however, the DRPO aims to guarantee reasonable performance for *all* distribution maps in an uncertainty collection around the nominal distribution map. To ensure the performance of the DRPO on a (set of) specific distribution map(s), the radius $\rho$ (or the tilt $\alpha$) must be calibrated. Lastly, each shaded region in figures below shows the curve's standard error of the mean from 30 trials.

## 5.1 Strategic Classification with Misspecified Cost Function

In reference to Example 2.1, we examine strategic classification involving a cost function that is misspecified. The experimental setup resembles that in [29]. The task is credit scoring, specifically predicting debt default. Individuals strategically manipulate their features to obtain a favorable classification.

Consider an instantiation of the response map (2.4) such that $u_\theta(x) = -\theta^\top x$ and $c(x, x') = \frac{1}{2\epsilon} \times \|x_{\mathrm{strat}} - x'_{\mathrm{strat}}\|_2^2 + \infty \times \|x_{\mathrm{non\text{-}strat}} - x'_{\mathrm{non\text{-}strat}}\|_2^2$. Without loss of generality, let the first $m$ features be strategic features and the last $d - m$ features be non-strategic features. Let $B = \mathrm{diag}(1, \ldots, 1, 0, \ldots, 0) \in \mathbb{R}^{d \times d}$ where the first $m$ diagonal elements are 1's and the others are 0's. Then the best response function is $\Delta_\theta(x) = x - \epsilon B\theta$.

For the base distribution $\mathcal{D}(\mathbf{0})$, we use a class-balanced subset of a Kaggle credit scoring dataset ([16], CC-BY 4.0). Features encompass an individual's historical data, including their monthly income and credit line count. Labels are binary where the value of 1 represents a default on a debt and 0 otherwise. There are a total of 3 strategic features and 6 non-strategic features. We use logistic model for the classifier and the cross-entropy loss with $L_2$-regularization for the loss function $\ell$.

Consider the cost function is misspecified by its performativity level $\epsilon$. We specify the cost function with the nominal performativity level $\epsilon = 0.5$. However, the true performativity level $\epsilon_{\mathrm{true}}$ might not be 0.5, but in the range of $[0.5 - 0.5\eta, 0.5 + 0.5\eta]$ for some $\eta \geq 0$.

The left plot of Figure 1 shows performative risk incurred by the PO and the DRPO's with various radius $\rho$'s. Note that the PO can be understood as the DRPO with $\rho = 0$. As $\rho$ increases, the DRPO aims to achieve more uniform performance across a wider range of $\epsilon_{\mathrm{true}}$. The middle plot of Figure 1 shows relative improvement in worst-case performative risk[1] of the DRPO to the PO as the radius $\rho$

---

[1] The relative worst-case improvement of $\theta$ to $\theta_{\mathrm{PO}}$ is $\frac{\max_{\epsilon_{\mathrm{true}}} \mathrm{PR}_{\mathrm{true}}(\theta_{\mathrm{PO}}) - \max_{\epsilon_{\mathrm{true}}} \mathrm{PR}_{\mathrm{true}}(\theta)}{\max_{\epsilon_{\mathrm{true}}} \mathrm{PR}_{\mathrm{true}}(\theta_{\mathrm{PO}})} \times 100\%$.

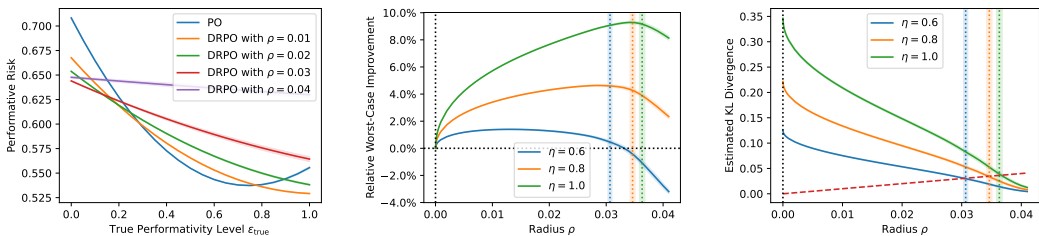

Figure 1: Results of Experiment 5.1. Left: performative risk incurred by the PO and the DRPO's with various radius $\rho$'s. Middle: relative improvement in worst-case performative risk of the DRPO to the PO as the radius $\rho$ increases, for different range of misspecification $\eta$'s. Right: radius $\rho$ versus estimated KL divergence between $\mathcal{D}_{\text{true}}(\theta_{\text{DRPO}}(\rho))$ and $\mathcal{D}(\theta_{\text{DRPO}}(\rho))$, where vertical bands indicate the calibrated radius $\rho_{\text{cal}}$'s.

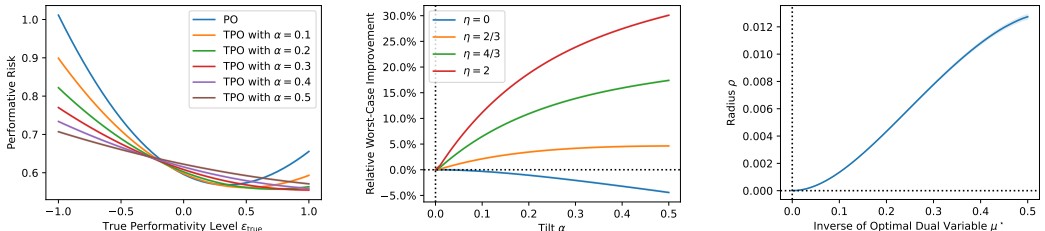

Figure 2: Results of Experiment 5.2. Left: performative risk incurred by the PO and the TPO's with various tilt $\alpha$'s. Middle: relative improvement in worst-case performative risk of the TPO to the PO as the tilt $\alpha$ increases, for different range of misspecification $\eta$'s. Right: the correspondence relationship between the radius $\rho$ and the (inverse of) optimal dual variable $\mu^\star$.

increases, for different range of misspecification $\eta$'s. Curves positioned above the horizontal dotted line indicate that the DRPO outperform the PO in terms of worst-case performative risk. When there is a larger range of misspecification $\eta$, the DRPO has greater potential to beat the PO. The right plot of Figure 1 demonstrates the post-fitting calibration of radius $\rho_{\text{cal}}$ described in Section 4.3. The vertical bands indicate the calibrated radius $\rho_{\text{cal}}$'s, which lead to the DRPO's with sound relative worst-case improvement and avoid overconservative solutions, as depicted by the corresponding bands in the middle plot of Figure 1.

## 5.2 Partially Identifiable Distribution Map

Recall Example 2.2, we examine a location model for distribution map where the misspecification arises from the estimation error of the model parameter. Let $V = [\theta_1 - \theta_0 \mid \theta_2 - \theta_0 \mid \cdots \mid \theta_K - \theta_0] \in \mathbb{R}^{d \times K}$ and $U = [\mu_1 - \mu_0 \mid \mu_2 - \mu_0 \mid \cdots \mid \mu_K - \mu_0] \in \mathbb{R}^{d \times K}$ where $d = \dim(\Theta)$. The unknown parameter $A$ can only be partially indentified through the equation $AV = U$ when $K < d$. A particular estimate of $A$ is $\widehat{A} = UV^\dagger = U(V^\top V)^{-1}V^\top$, where $V^\dagger$ is the Moore-Penrose inverse of $V$. In fact, the parameter $A$ is only identifiable up to the subspace $\mathcal{W} = \{UV^\dagger + E \mid \text{span}\{E^\top\} \subset \mathcal{N}(V^\top)\}$, where $\mathcal{N}(V^\top)$ is the left null space of $V$. Precicely, we have $AV = U$ if and only if $A \in \mathcal{W}$.

In this experiment, we still use the credit data. We observe sampling distribution of $\mathcal{D}(\mathbf{0})$, $\mathcal{D}(\mathbf{e}_1)$, and $\mathcal{D}(\mathbf{e}_2)$, where $\mathbf{e}_i$ is the $i$-th canonical basis. For the true distribution map, the performativity of the first two features is 0.5, while the performativity of the other 7 features is $\epsilon_{\text{true}}$. In short, $A_{\text{true}} = \text{diag}(0.5, 0.5, \epsilon_{\text{true}}, \ldots, \epsilon_{\text{true}})$. We set the range of $\epsilon_{\text{true}}$ to be $[-0.5\eta, 0.5\eta]$ for $\eta \geq 0$. By using the estiamte $\widehat{A}$, we model the performativity of the first two features correctly, but wrongly model the other features as non-strategic. This time we fit TPO by Algorithm 2 instead of DRPO.

The left plot of Figure 2 shows performative risk incurred by the PO and the TPO's with various tilt $\alpha$'s. As $\alpha$ increases, the TPO performs more uniformly across a wider range of $\epsilon_{\text{true}}$. The middle plot of Figure 2 shows relative improvement in worst-case performative risk of the TPO to the PO as the tilt $\alpha$ increases, for different range of misspecification $\eta$'s. Without misspeicification, $\eta = 0$, the PO is for sure better than the TPO. With moderate to large misspecification, $\eta \in \{2/3, 4/3, 2\}$, the TPO demonstrates certain advantages over the PO. The right of Figure 2 shows the relationship

between the distributionally robust performative risk minimization and the tilted performative risk minimization: fitting DRPO with $\rho$ (which returns the optimal dual variable $\mu^\star$) is equivalent to fitting TPO with $\alpha = 1/\mu^\star$.

## 5.3 Fairness without Demographics

Referencing to Example 2.3, we examine the scenario where the population distribution map is a mixture of two subpopulation distribution maps. We train a classifier using the population distribution map $\mathcal{D}_{\mathrm{pop}}$, but target at its performance on both the majority and minority, $\mathcal{D}_{\mathrm{maj}}$ and $\mathcal{D}_{\mathrm{min}}$. The distribution map is therefore misspecified due to subpopulation shift.

The experimental setup resembles that in [28]. Note that the credit dataset used in the previous experiments lacks demographic features. To enable oracle access to demographic information, synthetic data is generated for a performative classification task. The synthetic dataset exemplifies a scenario in which a linear decision boundary is not able to effectively classify both the majority and minority groups, necessitating a trade-off between them.

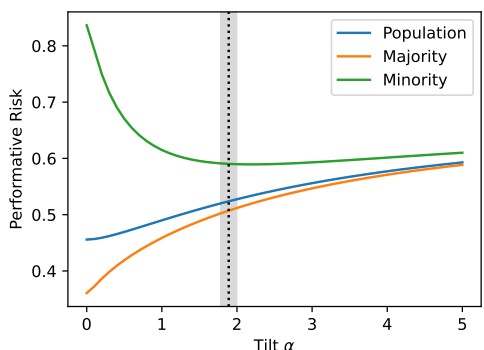

Figure 3: Results of Experiment 5.3. Performative risk of the population, the majority, and the minority, as the tilt $\alpha$ increases. The vertical band indicates the calibrated tilt $\alpha_{\mathrm{cal}}$'s.

Figure 3 shows performative risk of the population, the majority, and the minority incurred by the TPO, as the tilt $\alpha$ increases. The PO (which is TPO with $\alpha = 0$) exhibits the lowest performative risk at the population, but the greatest disparity between its performance for the majority and minority groups. As $\alpha$ increases, the TPO reduces the performance gap between the two groups at the expense of an increased population performative risk. This suggests that the distributionally robust performative prediction framework has the potential to mitigate unfairness towards the minority group, even in the absence of demographic information. Using a small calibration set with demographics, we can calibrate the tilt $\alpha_{\mathrm{cal}}$'s via the calibration set approach described in Section 4.3. The goal is to calibrate the tilt to satisfy a four-fifth rule[2] with minimal population performative risk. The vertical bands in Figure 3 shows the calibrated tilt $\alpha_{\mathrm{cal}}$'s reasonably meet the goal.

## 6 Summary and Discussion

In this work, we present a distributionally robust performative prediction framework that aims to improve robustness against a variety of distribution maps, thereby mitigating the problem of distribution map misspecification. We show provable guarantees for DRPO as a robust approximation to the true PO when the nominal distribution map differs from the actual one. We developed efficient algorithms for minimizing the distributionally robust performative risk. Empirical experiments are conducted to support our theoretical findings and validate the proposed algorithms.

The components of our approach are not new, but we are combining them in a novel way to solve a relevant problem. To be precise, the proposed approach is novel in the context to use the idea of distributional robustness to solve the practical problem of distribution map misspecification in performative prediction. In addition, it is novel to study the solution concept of distributionally robust performative optimum (DRPO), both theoretically and algorithmically.

In Appendix H, we extend the KL divergence distributionally robust performative prediction framework to a general $\varphi$-divergence distributionally robust performative prediction framework. Furthermore, it is possible to go beyond general $\varphi$-divergence. An extension to a Wasserstein DRO version is a natural direction for future research, calling for the development of new algorithms.[3]

---

[2]The minority's performative risk is not 25% higher than that of the majority, as motivated by the four-fifth rule documented in Uniform Guidelines on Employment Selection Procedures, 29 C.F.R. §1607.4(D) (2015).

[3]Due to space constraints, we provide additional materials for discussion (e.g., limitations) in Appendix I.

## Acknowledgements

This work was supported by the NSF under grants 2113364, 2113373, 2414918.

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

# Supplementary Materials for
# Distributionally Robust Performative Prediction

This supplementary materials contain the omitted details, technical proofs, and additional results pertaining to the main article "Distributionally Robust Performative Prediction". In Section A, the missing deriving steps for the toy example in Section 2.4 are provided. In Section B, we provide a characterization of the worst-case distribution map which attains the supremum in (2.5), and show that the DR performative prediction regulates the right tail of the performative losses. In Section C, we show a generalized excess risk bound for the DRPO. In Section D, all of the deferred proofs are presented. In Section E, we recall a standard algorithm for performative risk minimization. In Section F, we explain the claim in Section 4.2. In Section G, we provide omitted experimental details and additional empirical results. In Section H, we generalize the KL divergence DR performative prediction framework to a general $\varphi$-divergence DR performative prediction version, and propose an algorithm to find the associated DRPO. In Section I, we provide additional materials for discussion.

# A  Toy Example in Section 2.4 (Continued)

Recall that $\mathcal{Z} = \mathbb{R}$, $\Theta = [-1, 1]$, $\ell(z; \theta) = \theta z$, and the nominal distribution map is $\mathcal{D}(\theta) = \mathcal{N}(f(\theta), \sigma^2)$ for some $f : [-1, 1] \to \mathbb{R}$ and $\sigma^2 > 0$. Then the nominal performative risk is

$$\mathrm{PR}(\theta) = \mathbb{E}_{Z \sim \mathcal{N}(f(\theta), \sigma^2)}[\ell(Z; \theta)] = \theta f(\theta).$$

We firstly show that the distributionally robust performative risk is given by

$$\mathrm{DRPR}(\theta) = \mathrm{PR}(\theta) + \sqrt{\rho}\, \mathrm{Penalty}(\theta),$$

where the penalty function $\mathrm{Penalty}(\theta) = \sqrt{2\sigma^2}\, |\theta|$.

*Proof.* By Proposition 3.1, the strong duality of DRPR, and the well-established moment generating function of Gaussian distribution, we have

$$\begin{aligned}
\mathrm{DRPR}(\theta) &= \sup_{\widetilde{\mathcal{D}}:\widetilde{\mathcal{D}} \in \mathcal{U}(\mathcal{D})} \mathbb{E}_{Z \sim \widetilde{\mathcal{D}}(\theta)}[\ell(Z; \theta)] \\
&= \inf_{\mu \geq 0} \left\{ \mu \log \mathbb{E}_{Z \sim \mathcal{D}(\theta)} \left[ e^{\ell(Z;\theta)/\mu} \right] + \mu\rho \right\} \\
&= \inf_{\mu \geq 0} \left\{ \mu \left[ \frac{\theta f(\theta)}{\mu} + \frac{\sigma^2 \theta^2}{2\mu^2} \right] + \mu\rho \right\} \\
&= \theta f(\theta) + \inf_{\mu \geq 0} \left\{ \frac{\sigma^2 \theta^2}{2\mu} + \mu\rho \right\} \\
&= \theta f(\theta) + \sqrt{2\rho\sigma^2}\, |\theta| = \mathrm{PR}(\theta) + \sqrt{\rho}\, \mathrm{Penalty}(\theta).
\end{aligned}$$

Therefore, we derive an alternative form of $\mathrm{DRPR}(\theta)$, which is $\mathrm{PR}(\theta)$ with an $L_1$-regularization term. $\qquad \square$

Now we are in a more concrete setup that $f(\theta) = a_1\theta + a_0$ for some $a_1, a_0 > 0$. Recall that for any $\widetilde{\mathcal{D}} \in \mathcal{U}(\mathcal{D})$, we denote the performative risk of $\widetilde{\mathcal{D}}$ be

$$\mathrm{PR}_{\widetilde{\mathcal{D}}}(\theta) = \mathbb{E}_{Z \sim \widetilde{\mathcal{D}}(\theta)}[\ell(Z; \theta)].$$

We secondly show that

$$\sup_{\widetilde{\mathcal{D}} \in \mathcal{U}(\mathcal{D})} \mathrm{PR}_{\widetilde{\mathcal{D}}}(\theta_{\mathrm{PO}}) \geq \sup_{\widetilde{\mathcal{D}} \in \mathcal{U}(\mathcal{D})} \mathrm{PR}_{\widetilde{\mathcal{D}}}(\theta_{\mathrm{DRPO}}) + \frac{\rho\sigma^2}{2a_1}$$

for any fixed $\rho \leq \frac{a_0^2}{2\sigma^2}$.

*Proof.* In fact, we have

$$\theta_{\mathrm{PO}} = \frac{-a_0}{2a_1}, \theta_{\mathrm{DRPO}} = \frac{\sqrt{2\rho\sigma^2} - a_0}{2a_1}, \text{ and } \mathrm{DRPR}(\theta_{\mathrm{DRPO}}) = \frac{-(a_0 - \sqrt{2\rho\sigma^2})^2}{4a_1}.$$

By Proposition 2.6, the generalization principle of distributionally robust performative prediction, we have

$$\sup_{\widetilde{\mathcal{D}} \in \mathcal{U}(\mathcal{D})} \mathrm{PR}_{\widetilde{\mathcal{D}}}(\theta_{\mathrm{DRPO}}) \leq \mathrm{DRPR}(\theta_{\mathrm{DRPO}}) = \frac{-(a_0 - \sqrt{2\rho\sigma^2})^2}{4a_1}.$$

On the other hand, consider an *adversarial distribution map* $\mathcal{D}_{\mathrm{adv}}(\theta) = \mathcal{N}(a_1\theta + a_0 - \sqrt{2\rho\sigma^2}, \sigma^2)$. One can show that $D_{\mathrm{KL}}(\mathcal{D}_{\mathrm{adv}}(\theta), \mathcal{D}(\theta)) = \rho$ for all $\theta \in \Theta$, and therefore $\mathcal{D}_{\mathrm{adv}} \in \mathcal{U}(\mathcal{D})$. Then

$$\sup_{\widetilde{\mathcal{D}} \in \mathcal{U}(\mathcal{D})} \mathrm{PR}_{\widetilde{\mathcal{D}}}(\theta_{\mathrm{PO}}) \geq \mathrm{PR}_{\mathcal{D}_{\mathrm{adv}}}(\theta_{\mathrm{PO}}) = \frac{-(a_0 - \sqrt{2\rho\sigma^2})^2}{4a_1} + \frac{\rho\sigma^2}{2a_1}.$$

Therefore we complete the proof. $\qquad \square$

**Multi-variate case.** We extend the uni-variate example to multi-variate case. Let $\mathcal{Z} = \mathbb{R}^d$, $\Theta = [-1,1]^d$, and $\ell(z;\theta) = \theta^\top z$. Let the distribution map be $\mathcal{D}(\theta) = \mathcal{N}_d(f(\theta), \Sigma)$ for some $f : [-1,1]^d \to \mathbb{R}^d$ and $\Sigma$ is positive semi-definite. One can show that the performative risk is

$$\mathrm{PR}(\theta) = \mathbb{E}_{Z \sim \mathcal{D}(\theta)}[\ell(Z;\theta)] = \mathbb{E}_{Z \sim \mathcal{N}_d(f(\theta), \Sigma)}[\ell(Z;\theta)] = \theta^\top f(\theta)$$

and the distributionally robust performative risk is

$$\mathrm{DRPR}(\theta) = \sup_{\widetilde{\mathcal{D}}:\widetilde{\mathcal{D}} \in \mathcal{U}(\mathcal{D})} \mathbb{E}_{Z \sim \widetilde{\mathcal{D}}(\theta)}[\ell(Z;\theta)] = \theta^\top f(\theta) + \sqrt{2\rho\theta^\top \Sigma\theta} = \mathrm{PR}(\theta) + \sqrt{\rho}\,\mathrm{Penalty}(\theta),$$
(A.1)

where the penalty function $\mathrm{Penalty}(\theta) = \sqrt{2\theta^\top \Sigma\theta}$ penalizes the deviation of $\theta$ from the origin $0$, and the critical radius $\rho$ tunes the level of regularization. We also note that in this case the distributionally robust performative prediction problem can be transformed to a *second-order cone constrained program* (A.1).

# B    Worst-Case Distribution Map

As a byproduct of the proof of Proposition 3.1, here we provide a characterization of the worst-case distribution map which attains the supremum in (2.5).

**Proposition B.1** (Worst-case distribution map). *Suppose the problem* (3.1) *is solved by the unique* $\mu^\star(\theta) > 0$ *for any* $\theta \in \Theta$. *We define a distribution map* $\widetilde{\mathcal{D}}$ *satisfying the density ratio equation*

$$\frac{d\widetilde{\mathcal{D}}(\theta)}{d\mathcal{D}(\theta)} = \frac{e^{\ell(Z;\theta)/\mu^\star(\theta)}}{\mathbb{E}_{Z \sim \mathcal{D}(\theta)}[e^{\ell(Z;\theta)/\mu^\star(\theta)}]}, \quad \forall \theta \in \Theta.$$
(B.1)

*Then we have that* $\widetilde{\mathcal{D}}$ *is the unique worst-case distribution map, that is,*

$$\widetilde{\mathcal{D}}(\theta) = \arg\sup_{\widetilde{\mathcal{D}}(\theta):\widetilde{\mathcal{D}} \in \mathcal{U}(\mathcal{D})} \mathbb{E}_{Z \sim \widetilde{\mathcal{D}}(\theta)}[\ell(Z;\theta)].$$
(B.2)

Proposition B.1 shows that the worst-case distribution map $\widetilde{\mathcal{D}}$ is an *exponentially tilted* distribution map with respect to the nominal distribution map $\mathcal{D}$, where $\widetilde{\mathcal{D}}$ puts more weights on the high ends.

Figure 4 shows histogram of performative loss for the PO, the DRPO with $\rho = 0.02$, and the DRPO with $\rho = 0.04$, under the setup of Experiment 5.1 with $\epsilon_{\mathrm{true}} = 0.5$. These plots displayed in a left-to-right manner demonstrate that the DRPO regulates the right tail of the performative losses. Moreover, as the radius $\rho$ increases, there is a corresponding increase in the degree of regulation effects.

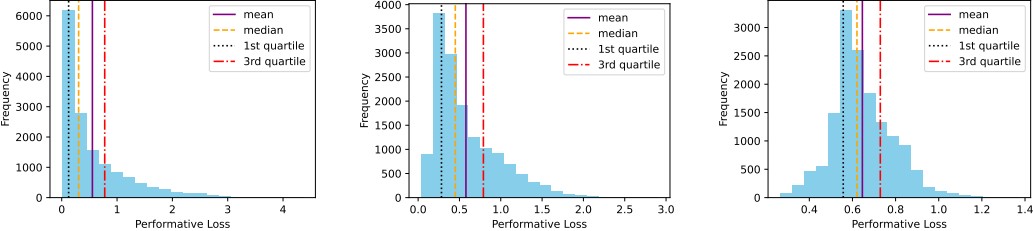

Figure 4: Histogram of performative loss under Experiment 5.1 with $\epsilon_{\mathrm{true}} = 0.5$. Left: histogram for the PO. Middle: histogram for the DRPO with $\rho = 0.02$. Right: hitogram for the DRPO with $\rho = 0.04$.

# C    Generalized Excess Risk Bound

We upper bound $\mathcal{E}(\theta_{\mathrm{DRPO}})$ when the uncertainty collection doesn't cover the true distribution map. To be specific, we analyze the excess risk bound of the DRPO when $D(\mathcal{D}_{\mathrm{true}}(\theta_{\mathrm{DRPO}}) \| \mathcal{D}(\theta_{\mathrm{DRPO}})) > \rho$.

**Proposition C.1** (Excess risk bound of the DRPO in general). *Suppose that we have bounded loss function $|\ell(z;\theta)| \leq B$ for any $z \in \mathcal{Z}, \theta \in \Theta$ and some $B > 0$. Then we have*

$$
\begin{aligned}
\mathcal{E}(\theta_{\mathrm{DRPO}}) \leq & \sqrt{\rho \operatorname{Var}_{Z \sim \mathcal{D}(\theta_{\mathrm{PO,true}})}[\ell(Z;\theta_{\mathrm{PO,true}})]} + o(\sqrt{\rho}) \\
& + \sqrt{2}B \inf_{P:D(P\|\mathcal{D}(\theta_{\mathrm{DRPO}})) \leq \rho} \sqrt{D(\mathcal{D}_{\mathrm{true}}(\theta_{\mathrm{DRPO}})\|P)}.
\end{aligned}
\tag{C.1}
$$

Comparing (C.1) to (3.3), we have an additional term in the excess risk bound which accommodates and accounts for the misspecification of uncertainty set around $\mathcal{D}(\theta_{\mathrm{DRPO}})$, which doesn't necessarily cover $\mathcal{D}_{\mathrm{true}}(\theta_{\mathrm{DRPO}})$. Furthermore, it is not difficult to see that if $D(\mathcal{D}_{\mathrm{true}}(\theta_{\mathrm{DRPO}})\|\mathcal{D}(\theta_{\mathrm{DRPO}})) \leq \rho$, then the last infimum term in the upper bound of (C.1) vanishes and (C.1) reduces to (3.3). Therefore, Proposition C.1 provides a generalized excess risk bound for the DRPO than Proposition 3.3.

# D    Deferred Proofs

For simplicity of notation, we denote the true PO by $\theta_{\mathrm{PO}}^{\star}$, which is denoted by $\theta_{\mathrm{PO,true}}$ in the main article.

## D.1    Proof of Proposition 3.1 and Proposition B.1

*Proof of Proposition 3.1.* We only have to show that for any fixed $\theta \in \Theta$, the dual form (3.1) holds. This follows Theorem 1 presented in [13]. □

We refer the readers to Proposition H.2 for a standard derivation for the dual form of a general $\varphi$-divergence distributionally robust performative risk.

*Proof of Proposition B.1.* This follows Proposition 1 presented in [13] and the discussion paragraph right after the proposition. □

## D.2    Proof of Proposition 3.2

*Proof.* We have the following decomposition of the excess risk bound of the PO,

$$
\begin{aligned}
& \mathrm{PR}_{\mathrm{true}}(\theta_{\mathrm{PO}}) - \min_{\theta \in \Theta} \mathrm{PR}_{\mathrm{true}}(\theta) \\
= & \mathrm{PR}_{\mathrm{true}}(\theta_{\mathrm{PO}}) - \mathrm{PR}_{\mathrm{true}}(\theta_{\mathrm{PO}}^{\star}) \\
= & \mathrm{PR}_{\mathrm{true}}(\theta_{\mathrm{PO}}) - \mathrm{PR}(\theta_{\mathrm{PO}}) + \underbrace{\mathrm{PR}(\theta_{\mathrm{PO}}) - \mathrm{PR}(\theta_{\mathrm{PO}}^{\star})}_{\leq 0 \text{ by definition of } \theta_{\mathrm{PO}}} + \mathrm{PR}(\theta_{\mathrm{PO}}^{\star}) - \mathrm{PR}_{\mathrm{true}}(\theta_{\mathrm{PO}}^{\star}) \\
\leq & 2 \sup_{\theta \in \Theta} |\mathrm{PR}(\theta) - \mathrm{PR}_{\mathrm{true}}(\theta)| \\
= & 2 \sup_{\theta \in \Theta} \left| \mathbb{E}_{Z \sim \mathcal{D}(\theta)}[\ell(Z;\theta)] - \mathbb{E}_{Z \sim \mathcal{D}_{\mathrm{true}}(\theta)}[\ell(Z;\theta)] \right| \\
\leq & 2B \sup_{\theta \in \Theta} D_{\mathrm{TV}}(\mathcal{D}_{\mathrm{true}}(\theta)\|\mathcal{D}(\theta)) \\
\leq & \sqrt{2}B \sup_{\theta \in \Theta} \sqrt{D_{\mathrm{KL}}(\mathcal{D}_{\mathrm{true}}(\theta)\|\mathcal{D}(\theta))}.
\end{aligned}
$$

The last two inequalities are due to the asumption of bounded loss, $|\ell(z;\theta)| \leq B$, and Pinksker's inequality. □

### D.3 Proof of Proposition 3.3

*Proof.* We have the following decomposition of the excess risk bound of the DRPO,

$$\text{PR}_{\text{true}}(\theta_{\text{DRPO}}) - \min_{\theta \in \Theta} \text{PR}_{\text{true}}(\theta)$$

$$= \text{PR}_{\text{true}}(\theta_{\text{DRPO}}) - \text{PR}_{\text{true}}(\theta_{\text{PO}}^\star)$$

$$= \underbrace{\text{PR}_{\text{true}}(\theta_{\text{DRPO}}) - \text{DRPR}(\theta_{\text{DRPO}})}_{\leq 0 \text{ by generalization principle}} + \underbrace{\text{DRPR}(\theta_{\text{DRPO}}) - \text{DRPR}(\theta_{\text{PO}}^\star)}_{\leq 0 \text{ by definition of } \theta_{\text{DRPO}}} + \text{DRPR}(\theta_{\text{PO}}^\star) - \text{PR}_{\text{true}}(\theta_{\text{PO}}^\star)$$

$$\leq \text{DRPR}(\theta_{\text{PO}}^\star) - \text{PR}_{\text{true}}(\theta_{\text{PO}}^\star)$$

$$= \sqrt{\text{Var}_{Z \sim \mathcal{D}(\theta_{\text{PO}}^\star)}[\ell(Z; \theta_{\text{PO}}^\star)]\rho} + o(\sqrt{\rho}).$$

The last equality is due to the sensitivity property of KL divergence-based DRO [9]. $\qquad\square$

### D.4 Proof of Proposition C.1

*Proof.* For any distribution $P$ such that $D_{\text{KL}}(P\|\mathcal{D}(\theta_{\text{DRPO}})) \leq \rho$, we have the following decomposition of the excess risk bound of the DRPO,

$$\text{PR}_{\text{true}}(\theta_{\text{DRPO}}) - \min_{\theta \in \Theta} \text{PR}_{\text{true}}(\theta)$$

$$= \text{PR}_{\text{true}}(\theta_{\text{DRPO}}) - \text{PR}_{\text{true}}(\theta_{\text{PO}}^\star)$$

$$= \text{PR}_{\text{true}}(\theta_{\text{DRPO}}) - \mathbb{E}_{Z \sim P}[\ell(Z; \theta_{\text{DRPO}})] + \underbrace{\mathbb{E}_{Z \sim P}[\ell(Z; \theta_{\text{DRPO}})] - \text{DRPR}(\theta_{\text{DRPO}})}_{\leq 0 \text{ by generalization principle}}$$

$$+ \underbrace{\text{DRPR}(\theta_{\text{DRPO}}) - \text{DRPR}(\theta_{\text{PO}}^\star)}_{\leq 0 \text{ by definition of } \theta_{\text{DRPO}}} + \text{DRPR}(\theta_{\text{PO}}^\star) - \text{PR}_{\text{true}}(\theta_{\text{PO}}^\star)$$

$$\leq \text{PR}_{\text{true}}(\theta_{\text{DRPO}}) - \mathbb{E}_{Z \sim P}[\ell(Z; \theta_{\text{DRPO}})] + \text{DRPR}(\theta_{\text{PO}}^\star) - \text{PR}_{\text{true}}(\theta_{\text{PO}}^\star)$$

$$= \text{PR}_{\text{true}}(\theta_{\text{DRPO}}) - \mathbb{E}_{Z \sim P}[\ell(Z; \theta_{\text{DRPO}})] + \sqrt{\text{Var}_{Z \sim \mathcal{D}(\theta_{\text{PO}}^\star)}[\ell(Z; \theta_{\text{PO}}^\star)]\rho} + o(\sqrt{\rho})$$

$$= \mathbb{E}_{Z \sim \mathcal{D}_{\text{true}}(\theta_{\text{DRPO}})}[\ell(Z; \theta_{\text{DRPO}})] - \mathbb{E}_{Z \sim P}[\ell(Z; \theta_{\text{DRPO}})] + \sqrt{\text{Var}_{Z \sim \mathcal{D}(\theta_{\text{PO}}^\star)}[\ell(Z; \theta_{\text{PO}}^\star)]\rho} + o(\sqrt{\rho})$$

$$\leq B D_{\text{TV}}(\mathcal{D}_{\text{true}}(\theta_{\text{DRPO}})\|P) + \sqrt{\text{Var}_{Z \sim \mathcal{D}(\theta_{\text{PO}}^\star)}[\ell(Z; \theta_{\text{PO}}^\star)]\rho} + o(\sqrt{\rho})$$

$$\leq \sqrt{2} B D_{\text{KL}}(\mathcal{D}_{\text{true}}(\theta_{\text{DRPO}})\|P) + \sqrt{\text{Var}_{Z \sim \mathcal{D}(\theta_{\text{PO}}^\star)}[\ell(Z; \theta_{\text{PO}}^\star)]\rho} + o(\sqrt{\rho}).$$

By the arbitrariness of $P$ in the divergence ball $D_{\text{KL}}(P\|\mathcal{D}(\theta_{\text{DRPO}})) \leq \rho$, we complete the proof. $\quad\square$

## E  Performative Risk Minimization

We recall a standard algorithm for performative risk minimization when the nominal distribution map $\mathcal{D}(\cdot)$ is modeled by a *response map* $T_\theta : \mathcal{Z} \to \mathcal{Z}$ and samples from the *base distribution* $\mathcal{D}_{\text{true}}(\mathbf{0})$:

$$Z \sim \mathcal{D}(\theta) \iff Z \overset{d}{=} T_\theta(Z_0) \text{ where } Z_0 \sim \mathcal{D}_{\text{true}}(\mathbf{0}).$$

This is a popular model for distribution map including strategic classification (see Example 2.1) and location family (see Example 2.2) as prominent examples.

It is not hard to see that the nominal distribution map $\mathcal{D}(\cdot)$ is fully characterized by the response map $T_\theta$ and the base measure $\mathcal{D}_{\text{true}}(\mathbf{0})$ because

$$\mathcal{D}(\theta) \overset{d}{=} T_{\theta\sharp}(\mathcal{D}_{\text{true}}(\mathbf{0})) \text{ for any } \theta \in \Theta,$$

that is, the measure $\mathcal{D}(\theta)$ is the *pushforward measure* of $\mathcal{D}_{\text{true}}(\mathbf{0})$ under the *transport map* $T_\theta$. The performative risk can then be reformulated as

$$\text{PR}(\theta) = \mathbb{E}_{Z \sim \mathcal{D}(\theta)}[\ell(Z; \theta)] = \mathbb{E}_{Z \sim T_{\theta\sharp}(\mathcal{D}_{\text{true}}(\mathbf{0}))}[\ell(Z; \theta)] = \mathbb{E}_{Z \sim \mathcal{D}_{\text{true}}(\mathbf{0})}[\ell(T_\theta(Z); \theta)].$$

With the last formula, the performative risk minimization problem becomes a standard (possibly nonconvex) stochastic optimization problem, which can be solved efficiently by any popular stochastic

optimization algorithm, *e.g.*, sample average approximation (SAA, [19]). given the loss function $\ell(z; \theta)$ and the response map $T_\theta(z)$ are sufficiently differentiable with respect to $\theta$.

We refer to Section 1.1 for a summary of existing performative risk minimization algorithms.

# F An Explanation to the Claim in Section 4.2

In Section 4.2, we claim that $\arg\min_{\theta \in \Theta} \{\psi(\theta, \mu)\} \approx \arg\min_{\theta \in \Theta} \{\mathbb{E}_{Z \sim \mathcal{D}(\theta)}[\ell(Z; \theta)]\}$ as $\mu \to \infty$. This is due to the fact of that $e^x \sim x + 1$ for small $x \to 0$ and the argmax/argmin theorem. Therefore, we have

$$\arg\min_{\theta \in \Theta} \left\{ \mu \log \mathbb{E}_{Z \sim \mathcal{D}(\theta)} \left[ e^{\ell(Z; \theta)/\mu} \right] + \mu\rho \right\}$$

$$\approx \arg\min_{\theta \in \Theta} \left\{ \mu \log \mathbb{E}_{Z \sim \mathcal{D}(\theta)} \left[ \frac{\ell(Z; \theta)}{\mu} + 1 \right] + \mu\rho \right\}$$

$$\approx \arg\min_{\theta \in \Theta} \left\{ \mathbb{E}_{Z \sim \mathcal{D}(\theta)}[\ell(Z; \theta)] \right\}$$

as $\mu \to \infty$.

# G Experimental Details and More Results

For the loss function $\ell(x, y; \theta)$, we adopt the cross-entropy loss with $L_2$-regularization, that is,

$$\ell(x, y; \theta) = -y \log h_\theta(x) - (1 - y) \log(1 - h_\theta(x)) + \frac{\lambda}{2} \|\theta\|_2^2,$$

where $h_\theta(x) = \left(1 + \exp\{-\theta^\top x\}\right)^{-1}$ and $\lambda = 0.001$. Given a model $\theta$, we predict $\widehat{y} = \mathbf{1}\{h_\theta(x) \geq 0.5\}$. In addition, each replicate of the experiments is run on a local machine with an Intel Xeon Gold 6154 CPU and 32GB of RAM in less than an hour execution time.

## G.1 Strategic Classification with Misspecified Cost Function in Section 5.1 (Continued)

The data preprocessing procedure for the credit dataset [16] follows the procedure documented in [29]. After that procedure, the base distribution $\mathcal{D}_{\text{true}}(\mathbf{0})$ has $n = 14878$ data points with equal probability mass. We treat the distribution map associated with $\mathcal{D}_{\text{true}}(\mathbf{0})$ as the underlying test distribution map which is unknown to us. We generate $n$ IID samples from $\mathcal{D}_{\text{true}}(\mathbf{0})$ to get a training base distribution $\widehat{\mathcal{D}}(\mathbf{0})$, that is, $\widehat{\mathcal{D}}(\mathbf{0}) \sim \mathcal{D}_{\text{true}}(\mathbf{0})^{\otimes n}$, and then we have a training nominal distribution map induced by $\widehat{\mathcal{D}}(\mathbf{0})$. This training procedure is repeated for 10 trials, and the shaded region in each of the figures in this paper represents the standard error of the mean calculated from the 10 trials for the corresponding curve.

The left of Figure 5 shows performative balanced error rate, which refers to the balanced error rate (BER) on the test model-induced distribution, incurred by the PO and the DRPO's with various radius $\rho$'s. Because the cross-entropy loss serves as a surrogate for classification error, we see patterns of these curves similar to those in the left of Figure 1: as $\rho$ increases, the DRPO achieves more uniform performance across a wider range of $\epsilon_{\text{true}}$. On the other hand, because the performative classification error is not exactly the criterion we minimize, different patterns are also observed: the DRPO with relatively large radius $\rho = 0.04$ outperforms the PO for all $\epsilon_{\text{true}} \in [0, 1]$. As indicated by the right of Figure 5, by increasing $\rho$ the DRPO constantly improves the relative worst-case performance in terms of performative BER for $\eta \in \{0.6, 0.8, 1.0\}$. Here the relative worst-case improvement in performative BER of $\theta$ to $\theta_{\text{PO}}$ is defined by $\frac{\max_{\epsilon_{\text{true}}} \text{BER}_{\text{true}}(\theta_{\text{PO}}) - \max_{\epsilon_{\text{true}}} \text{BER}_{\text{true}}(\theta)}{\max_{\epsilon_{\text{true}}} \text{BER}_{\text{true}}(\theta_{\text{PO}})} \times 100\%$.

For implementing the post-fitting calibration procedure in Section 4.3, we have to estimate the KL divergence between $\mathcal{D}_{\text{true}}(\theta_{\text{DRPO}}(\rho))$ and $\mathcal{D}(\theta_{\text{DRPO}}(\rho))$ for any fixed $\rho$. Here we index $\theta_{\text{DRPO}}$ by the radius $\rho$ to emphasize its dependence on $\rho$. Recall that

$$Z \sim \mathcal{D}(\theta) \iff Z \stackrel{d}{=} Z_0 - 0.5A\theta \text{ and } Z \sim \mathcal{D}_{\text{true}}(\theta) \iff Z \stackrel{d}{=} Z_0 - \epsilon_{\text{true}} A\theta \text{ where } Z_0 \sim \mathcal{D}_{\text{true}}(\mathbf{0}).$$

Here $A^\top = \begin{bmatrix} B^\top & | & \mathbf{0}_d \end{bmatrix}$ (see Section 5.2 for the definition of $B$) and we have training base distribution $\widehat{\mathcal{D}}(\mathbf{0})$ from $\mathcal{D}_{\text{true}}(\mathbf{0})$. We use an inexact parametric method to estimate the KL divergence. Assume

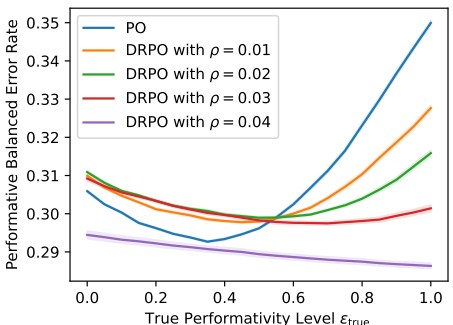
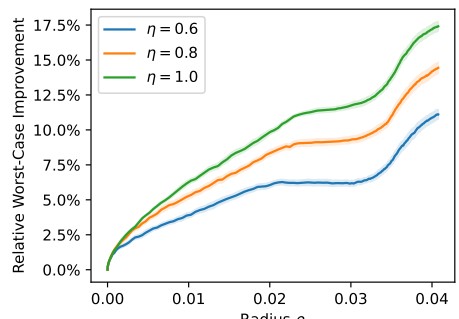

Figure 5: Additional results of Experiment 5.1. Left: performative balanced error rate incurred by the PO and the DRPO's with various radius $\rho$'s. Right: relative improvement in worst-case performative balanced error rate of the DRPO to the PO as the radius $\rho$ increases, for different range of misspecification $\eta$'s.

$\mathcal{D}_{\text{true}}(\mathbf{0}) \sim \mathcal{N}(\mu, \Sigma)$, then

$$D(\mathcal{D}_{\text{true}}\|\mathcal{D}) = \mathcal{D}(\mathcal{N}(\mu - 0.5A\theta_{\text{DRPO}}, \Sigma)\|\mathcal{N}(\mu - \epsilon_{\text{true}}A\theta_{\text{DRPO}}, \Sigma))$$
$$= 0.5(\epsilon_{\text{true}} - 0.5)^2\theta_{\text{DRPO}}^\top A^\top \Sigma^{-1} A\theta_{\text{DRPO}}.$$

Let $\widehat{\Sigma}$ be the sample covariance of $\widehat{D}(\mathbf{0})$, a plug-in method implies an estimate of the KL divergence, which is given by

$$\widehat{D}(\mathcal{D}_{\text{true}}(\theta_{\text{DRPO}}(\rho))\|\mathcal{D}(\theta_{\text{DRPO}}(\rho))) = 0.5(\epsilon_{\text{true}} - 0.5)^2\theta_{\text{DRPO}}^\top A^\top \widehat{\Sigma}^{-1} A\theta_{\text{DRPO}}.$$

### G.2 Partially Identifiable Distribution Map in Section 5.2 (Continued)

The data generating procedure is the same as that in Experiment 5.1. The left of Figure 6 shows performative balanced error rate (BER) incurred by the PO and the TPO's with various tilt $\alpha$'s. Similar to the left of Figure 5, we observe that 1) as $\alpha$ increases, the TPO achieves more uniform performance across a wider range of $\epsilon_{\text{true}}$; 2) the TPO with relatively large tilt $\alpha = 0.5$ outperforms the PO for all $\epsilon_{\text{true}} \in [-1, 1]$. The right of Figure 6 shows relative improvement in worst-case performative balanced error rate of the TPO to the PO as the tilt $\alpha$ increases, for different range of misspecification $\eta$'s. Without misspeicification, $\eta = 0$, the TPO has comparable performance to the PO in terms of performative BER. With moderate to large misspecification, $\eta \in \{2/3, 4/3, 2\}$, the TPO shows significant advantages over the PO in terms of relative improvement in worst-case performative BER.

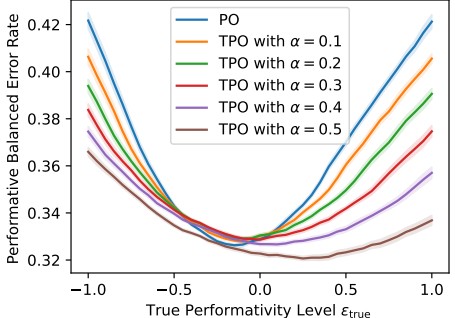
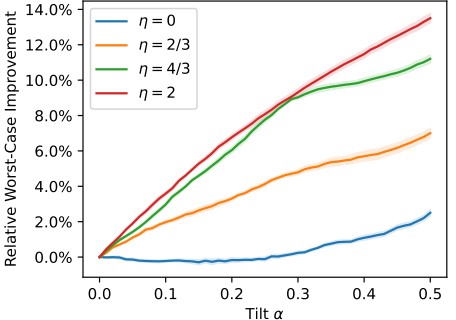

Figure 6: Additional results of Experiment 5.2. Left: performative balanced error rate incurred by the PO and the TPO's with various tilt $\alpha$'s. Right: relative improvement in worst-case performative balanced error rate of the TPO to the PO as the tilt $\alpha$ increases, for different range of misspecification $\eta$'s.

### G.3 Fairness without Demographics in Section 5.3 (Continued)

We adopt the following data generating process similar to that in [28]. Let $X \sim \gamma \mathcal{N}(\mu_A, \Sigma_A) + (1-\gamma)\mathcal{N}(\mu_B, \Sigma_B)$. Let $\gamma = 0.8$ so that group $A$ is the majority group and group $B$ is the minority group. Let $\mu_A = 1 \times \mathbf{1}_d$, $\mu_B = 0.8 \times \mathbf{1}_d$, and $\Sigma_A = \Sigma_B = 0.1 \times I_d$. If $X$ comes from group $A$, then label $Y = 0$ if $X^\top \mathbf{1}_d \leq \mu_A^\top \mathbf{1}_d$. If $X$ comes from group $B$, then label $Y = 0$ if $X^\top \mathbf{1}_d \leq \mu_B^\top \mathbf{1}_d$. The distribution map follows:

$$X_{1:\lfloor d/2 \rfloor} \leftarrow X_{1:\lfloor d/2 \rfloor} - \epsilon \theta_{1:\lfloor d/2 \rfloor}$$
$$X_{(\lfloor d/2 \rfloor+1):d} \leftarrow X_{(\lfloor d/2 \rfloor+1):d}$$

so that the first $\lfloor d/2 \rfloor$ features are strategic features, and $\epsilon$ controls the strength of performativity. We choose $d = 10$ and $\epsilon = 0.5$ to wrap up the setup. Finally, we assume knowledge of the true performativity and observe IID samples of size $n = 12500$ from the population base distribution. In short, we eliminate the effect of population distribution map misspecification, and instead concentrate on the effect of subpopulation distribution map shift.

Figure 7 shows the performative accuracy, which refers to the accuracy on the model-induced distribution, of the population, the majority, and the minority incurred by the TPO, as the tilt $\alpha$ increases. Because the cross-entropy loss serves as a surrogate for classification error, we see patterns of the three curves similar to those in Figure 3. The PO (which is TPO with $\alpha = 0$) exhibits the highest performative accuracy at the population, but the greatest disparity between its performance for the majority and minority groups. As $\alpha$ increases, the TPO reduces the performance gap between the two groups at the expense of an decreased population performative accuracy. This suggests that the distributionally robust performative prediction framework has the potential to mitigate unfairness towards the minority group, even in the absence of demographic information.

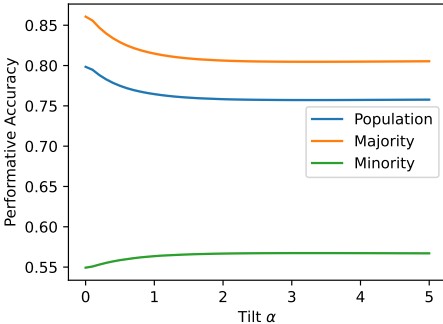

Figure 7: Additional results of Experiment 5.3. Performative accuracy of the population, the majority, and the minority, as the tilt $\alpha$ increases.

For implementing the calibration set procedure in Section 4.3, we have to obtain access to a few samples with the group membership information. Here we allow ourselves to observe IID samples of size $n_{\text{cal}} = 125$ with the known group membership from the population base distribution.

## H  Extension to General $\varphi$-Divergence

The KL divergence distributionally robust performative prediction framework can be extended to a general $\varphi$-divergence distributionally robust performative prediction framework. Recall that a $\varphi$-divergence is defined by

$$D_\varphi(Q \| P) = \int \varphi\left(\frac{dQ}{dP}\right) dP,$$

where $\varphi : \mathbb{R}_+ \to \mathbb{R}_+$ and $\varphi(1) = 0$. Here $dQ/dP$ is the Radon–Nikodym derivative, and we implicitly require the probability measure $Q$ to be absolutely continuous with respect to $P$. Note that if we choose $\varphi(t) = t \log t$, then we recover the framework presented in the main article. Now we keep $\varphi$ as a generic function and we will instantiate some popular families of $\varphi$-divergence after the presentation of the general framework.

With the $\varphi$-divergence, we can define a family of distribution maps around the nominal distribution map. Specifically, the *uncertainty collection* around $\mathcal{D}$ with *critical radius* $\rho$ is defined as

$$\mathcal{U}(\mathcal{D}) = \{\widetilde{\mathcal{D}} : \Theta \to \mathcal{P}(\mathcal{Z}) \mid D_\varphi(\widetilde{\mathcal{D}}(\theta)\|\mathcal{D}(\theta)) \leq \rho, \forall \theta \in \Theta\}.$$

**Definition H.1** ($\varphi$-divergence distributionally robust performative risk)**.** *The $\varphi$-divergence distributionally robust performative risk with the uncertainty collection $\mathcal{U}(\mathcal{D})$ is defined as*

$$\mathrm{DRPR}_\varphi(\theta) = \sup_{\widetilde{\mathcal{D}}:\widetilde{\mathcal{D}}\in\mathcal{U}(\mathcal{D})} \mathbb{E}_{Z\sim\widetilde{\mathcal{D}}(\theta)}[\ell(Z;\theta)]. \tag{H.1}$$

The evaluation of $\varphi$-divergence distributionally robust performative risk $\mathrm{DRPR}_\varphi(\theta)$ given in (H.1) involves an infinite dimensional maximization problem which is generally intractable. Fortunately, it is in fact equivalent to a minimization problem over two dual variables. This is given by the following strong duality result.

**Proposition H.2** (Strong duality of $\mathrm{DRPR}_\varphi$)**.** *For any $\theta \in \Theta$, we have*

$$\mathrm{DRPR}_\varphi(\theta) = \inf_{\mu\geq 0,\nu\in\mathbb{R}} \left\{ \mathbb{E}_{Z\sim\mathcal{D}(\theta)} \left[ \mu\varphi^* \left( \frac{\ell(Z;\theta)-\nu}{\mu} \right) + \mu\rho + \nu \right] \right\}, \tag{H.2}$$

*where $\varphi^*(s) = \sup_t\{st - \varphi(t)\}$ is the convex conjugate of $\varphi$.*

*Proof.* We only have to show that for any fixed $\theta \in \Theta$, the dual form (H.2) holds. We introduce the likelihood ratio $L(Z) = d\widetilde{\mathcal{D}}(\theta)/d\mathcal{D}(\theta)$. By change of variable, we can rewrite the $\varphi$-divergence distributionally robust performative risk (H.1) as

$$\sup_{\widetilde{\mathcal{D}}(\theta):D_\varphi(\widetilde{\mathcal{D}}(\theta)\|\mathcal{D}(\theta))\leq\rho} \mathbb{E}_{Z\sim\widetilde{\mathcal{D}}(\theta)}[\ell(Z;\theta)]$$
$$= \sup_{L\geq 0} \left\{ \mathbb{E}_{Z\sim\mathcal{D}(\theta)}[L(Z)\ell(Z;\theta)] \mid \mathbb{E}_{Z\sim\mathcal{D}(\theta)}[\varphi(L(Z))] \leq \rho, \mathbb{E}_{Z\sim\mathcal{D}(\theta)}[L(Z)] = 1 \right\},$$

where the supremum takes over measurable functions. This gives us a constrained optimization problem. Let $\mu \geq 0$ be the Lagrange multiplier for $\mathbb{E}_{Z\sim\mathcal{D}(\theta)}[\varphi(L(Z))] \leq \rho$ and $\nu \in \mathbb{R}$ be the Lagrange multiplier for $\mathbb{E}_{Z\sim\mathcal{D}(\theta)}[L(Z)] = 1$. The corresponding Lagrangian is

$$\mathcal{L}(L,\mu,\nu) = \mathbb{E}_{Z\sim\mathcal{D}(\theta)}[(\ell(Z;\theta)-\nu)L(Z) - \mu\varphi(L(Z))] + \mu\rho + \nu.$$

For regular $\varphi$-divergence, we have

$$\sup_{\widetilde{\mathcal{D}}(\theta):D_\varphi(\widetilde{\mathcal{D}}(\theta)\|\mathcal{D}(\theta))\leq\rho} \mathbb{E}_{Z\sim\widetilde{\mathcal{D}}(\theta)}[\ell(Z;\theta)]$$
$$= \inf_{\mu\geq 0,\nu\in\mathbb{R}} \sup_{L\geq 0} \mathcal{L}(L,\mu,\nu)$$
$$= \inf_{\mu\geq 0,\nu\in\mathbb{R}} \sup_{L\geq 0} \left\{ \mathbb{E}_{Z\sim\mathcal{D}(\theta)}[(\ell(Z;\theta)-\nu)L(Z) - \mu\varphi(L(Z))] + \mu\rho + \nu \right\}$$
$$= \inf_{\mu\geq 0,\nu\in\mathbb{R}} \sup_{L\geq 0} \left[ \mathbb{E}_{Z\sim\mathcal{D}(\theta)} \left[ \mu \left\{ \frac{\ell(Z;\theta)-\nu}{\mu}L(Z) - \varphi(L(Z)) \right\} + \mu\rho + \nu \right] \right]$$
$$= \inf_{\mu\geq 0,\nu\in\mathbb{R}} \left\{ \mathbb{E}_{Z\sim\mathcal{D}(\theta)} \left[ \mu\sup_{t\geq 0} \left\{ \frac{\ell(Z;\theta)-\nu}{\mu}t - \varphi(t) \right\} + \mu\rho + \nu \right] \right\}$$
$$= \inf_{\mu\geq 0,\nu\in\mathbb{R}} \left\{ \mathbb{E}_{Z\sim\mathcal{D}(\theta)} \left[ \mu\varphi^* \left( \frac{\ell(Z;\theta)-\nu}{\mu} \right) + \mu\rho + \nu \right] \right\}.$$

Here the last equality holds according to the definition of the convex conjugate $\varphi^*(\cdot)$. □

To develop an algorithm for finding the DRPO, we introduce the $(\mu,\nu)$-*augmented performative risk*.

**Definition H.3** ($(\mu,\nu)$-augmented performative risk)**.** *The $(\mu,\nu)$-augmented performative risk is defined by*

$$\mathrm{AugPR}_\varphi(\theta,\mu,\nu) = \mathbb{E}_{Z\sim\mathcal{D}(\theta)} \left[ \mu\varphi^* \left( \frac{\ell(Z;\theta)-\nu}{\mu} \right) + \mu\rho + \nu \right]. \tag{H.3}$$

---

**Algorithm 3** Augmented Performative Risk Minimization

---
1: **Input:** nominal distribution map $\mathcal{D}(\theta)$
2: Update $(\theta, \mu, \nu) \leftarrow \arg\min_{\theta \in \Theta, \mu \geq 0, \nu \in \mathbb{R}} \left\{ \mathbb{E}_{Z \sim \mathcal{D}(\theta)} \left[ \mu\varphi^* \left( \frac{\ell(Z;\theta) - \nu}{\mu} \right) + \mu\rho + \nu \right] \right\}$
3: **Return:** $\theta$

---

From (H.2) and (H.3), we see that the minimization problem of $\mathrm{DRPR}_\varphi(\theta)$ over $\theta$ is equivalent to the minimization problem of $\mathrm{AugPR}_\varphi(\theta, \mu, \nu)$ jointly over $(\theta, \mu, \nu)$, which is itself a performative risk minimization problem. We summarize this procedure in Algorithm 3.

For special choice of $\varphi$, the strong duality result (H.2) can be reduced to involving only a single dual variable. Now we instantiate $\varphi$-divergence as the Cressie-Read family:

$$\phi_k(t) = \frac{t^k - kt + k - 1}{k(k-1)},$$

for $k > 1$. Like KL divergence, a distributionally robust performative prediction problem induced by a divergence from the Cressie-Read family has a dual reformualtion with single dual variable:

$$\mathrm{DRPR}_{\phi_k}(\theta) = \inf_{\mu \geq 0} \left\{ (1 + \rho k(k-1))^{1/k} \mathbb{E}_{Z \sim \mathcal{D}(\theta)} \left[ \max\{\ell(Z;\theta) - \mu, 0\}^{k_*} \right]^{1/k_*} + \mu \right\}, \quad \text{(H.4)}$$

where $k_*$ is the conjugate number of $k$ such that $1/k + 1/k_* = 1$. Therefore, an algorithm parallel to Algirthm 1 can be developed in a similar fashion based on the single-variable dual form (H.4).

As a final remark, all of the theoretical results regarding excess risk bounds (see Proposition 3.3 and Proposition C.1) are still valid for the general $\varphi$-divergence distributionally robust performative prediction (which means the result statements won't change if one replaced the KL-divergence by any $\varphi$-divergence).

On the other hand, it is possible to extend Proposition 3.2 to general $\varphi$-divergence, and the result statement needs a slight modification. By generalized Pinsker's inequality, there is an increasing function $F : [0, 2] \to \mathbb{R}_+$ such that $D_\varphi(P\|Q) \geq F(\mathrm{TV}(P\|Q))$. Then for general $\varphi$-divergence, Proposition 3.2 can be modified to $\mathcal{E}(\theta_{\mathrm{PO}}) \leq 2B \sup_{\theta \in \Theta} F^{-1}(D_\varphi(\mathcal{D}_{\mathrm{true}}(\theta)\|\mathcal{D}(\theta))$ or $\mathcal{E}(\theta_{\mathrm{PO}}) \leq 2BF^{-1}(\sup_{\theta \in \Theta} D_\varphi(\mathcal{D}_{\mathrm{true}}(\theta)\|\mathcal{D}(\theta))$ due to monotonicity of $F(\cdot)$. For a concrete example, $F(v) = v^2 \mathbf{1}\{v < 1\} + \frac{v}{2-v} \mathbf{1}\{v \geq 1\}$ when $\varphi(t) = (t-1)^2$.

# I  Additional Discussion

In this appendix, we provide additional materials for discussion.

**Extension to Wasserstein distance.** It is possible to use Wasserstein distance to define the uncertainty collection within our framework. Our algorithms can be modified to compute Wasserstein DRPO. For example, one can establish the strong duality of Wasserstein DRPO and develop an alternating minimization algorithm similar to Algorithm 1. However, the new algorithm involves an additional step of transport cost-regularized loss maximization due to the more complex dual form of Wasserstein DRO. For the corresponding theory, we expect that the square root of variance in (3.3) would be replaced by the Lipschitz norm of $\ell(\cdot, \theta_{\mathrm{PO,true}})$.

**Conservativeness of $\rho_{\mathrm{cal}}$ and trade-off in selecting $\rho$.** The main text covers the discussion of the conservativeness of $\rho_{\mathrm{cal}}$ in two ways. Firstly, we show the trade-off in selecting $\rho$. For values of $\rho$ ranging from small to moderate, DRPO outperforms PO in terms of performative risk (and similarly for worst-case performative risk). Conversely, for large values of $\rho$, PO is better than DRPO. There exists an "sweet spot" of $\rho$ where DRPO yields maximal benefits over PO. This trade-off between DRPO and PO is demonstrated in any "vertical slices" of the left plot of Figure 1 (and similarly in the lines of the middle plot of Figure 1 for worst-case performance). Secondly, we show the performance of the calibrated radius $\rho_{\mathrm{cal}}$ in the middle plot of Figure 1, where the vertical bands indicate the calibrated radius $\rho_{\mathrm{cal}}$. Although $\rho_{\mathrm{cal}}$ doesn't achieve the best possible worst-case improvement (which is an impossible oracle), it achieves a comparable performance, especially when $\eta \in \{0.8, 1.0\}$. This demonstrates the effectiveness of $\rho_{\mathrm{cal}}$ chosen by our calibration method.

**Difference between TPO and TERM, and difference between DRPO and simple DRO.** TPO (see Section 4.2 and Algorithm 2) differs from TERM [22] in that it takes into account performativity,

whereas TERM ignores. To be precise, TPO minimizes $\mathbb{E}_{Z \sim \mathcal{D}(\theta)}[e^{\alpha \ell(Z;\theta)}]$ while TERM minimizes $\mathbb{E}_{Z \sim \mathcal{D}(\mathbf{0})}[e^{\alpha \ell(Z;\theta)}]$, where $\mathcal{D}(\theta)$ is the distribution map and $\mathcal{D}(\mathbf{0})$ is the base distribution. An analogous explanation applies to the difference between DRPO and simple DRO. TERM (or implicitly equivalently simple DRO) is ineffective in the context of this work's problem setup because it fails to account for performativity. Note that our methods are not doing DRO because the uncertainty set here depends on the model parameter $\theta$. We borrow the idea of distributional robustness, but we have a fundamentally different problem at hand.

**Absence of small calibration set.** The calibration set is not always available. Here we clarify the calibration set method and briefly discuss a possible solution when there is no calibration set. In the experiment of Section 5.3, we only need a small set of calibration data, which aligns with the regime of "weak group information". It is generally difficult to calibrate radius for uncertainty set and most distributional robustness related work only concerns the effect of increasing $\alpha$ and $\rho$. We do more to demonstrate that there are some practical calibration methods that work. In the absence of any calibration data, one can specify the radius by some prior belief. For example, consider $\mathcal{D}_{\mathrm{pop}}(\theta) = \gamma \mathcal{D}_{\mathrm{major}}(\theta) + (1 - \gamma)\mathcal{D}_{\mathrm{minor}}(\theta)$ as in Example 2.3. If one believes that $\gamma_1 \leq \gamma \leq \gamma_2$ for some $0 < \gamma_1, \gamma_2 < 1$, then one can upper bound the divergence $\mathcal{D}(\mathcal{D}_{\mathrm{major}}(\theta) \| \mathcal{D}_{\mathrm{pop}}(\theta)) \leq -\log(\gamma_1)$ and $\mathcal{D}(\mathcal{D}_{\mathrm{minor}}(\theta) \| \mathcal{D}_{\mathrm{pop}}(\theta)) \leq -\log(1 - \gamma_2)$. Further, one can choose $\rho = \max\{-\log(\gamma_1), -\log(1 - \gamma_2)\}$ when using DRPO. However, this choice of radius could be conservative.

**Algorithmic convergence.** The convergence guarantees of the proposed algorithms can be established on a case-by-case basis, depending on the specific "inner" algorithm or solver employed for solving performative risk minimization. Existing algorithmic convergence results for model-based PO solvers [26, 14] can be taped into our algorithms naturally. Moreover, empirical results in Section 5 validate the effectiveness of our proposed algorithms.

**Broader impacts.** This paper presents work whose goal is to advance the field of machine learning, especially the subfield of performative prediction. Despite the development of new algorithms, their direct societal impact are limited to those outlined in the original paper on performative prediction [29].

