# OpenReview forum: "Distributionally Robust Performative Prediction"
_NeurIPS.cc/2024/Conference — NeurIPS 2024 poster_

### Official Review · Reviewer_67u7 · 2024-07-11

**Soundness:** 4
**Presentation:** 4
**Contribution:** 4
**Rating:** 8
**Confidence:** 5

**Summary:**

In this paper, the authors propose a new framework and algorithm for finding performatively optimal solutions in the performative prediction framework. Performative optima are in general hard to find, and to do so, a number of approaches start by trying to estimate the underlying distribution map: a task which incurs fundamental statistical or modeling error. The main contribution of the paper is that the authors bring to bear tools from the robust optimization literature to find solutions which explicitly account for this error and are provably robust to it. The main technical results of the paper are structural theorems bounding the performance of the distributional robust performative optima, and guidelines around the selection of hyperparameters for their algorithm.

Disclosure: I previously reviewed this paper for ICML and I was disappointed that it was not accepted to the conference then. I thought it was a very strong paper back then and I think the authors have done an even better job now.

**Strengths:**

The motivation and ideas behind the paper are excellent. I think it’s an extremely natural solution to think about robust optimization approaches for finding a performatively optimal solution. The author provide very clear and insightful examples showing the value of their ideas and I’m glad they’ve established a bridge between these previously disparate areas of ML research. I think this paper is a significant contribution to the growing literature on performative prediction and will lead others to build on their work. The experimental section of the paper was particularly well executed and nicely rounds out the paper.

**Weaknesses:**

The paper has very few weaknesses, but to be somewhat nitpicky, I think that the authors would do well to clarify the computational complexity behind some of their algorithms and make precise statements regarding when and why the robust optimization problem can be solved, assuming that the non-robust (nominal) performative optimization problem is tractable. Please see below for further comments.

**Questions:**

“L77: it is easy to see..” Aren’t there cases where the optimization problem is still computationally, if not statistically, intractable?

Assume that PR(theta) = E_{z~ D(theta)} ell(z, theta) is a tractable optimization problem. I understand that the mapping x —> exp(x) is a monotonic transformation, but could you provide a formal statement showing why one would still be able to tractably solve this modified version of the problem, assuming that the first version is tractable?

Could you spell out in more detail exactly what the grid search is over when you state that L270 “for general problems, these calibration methods necessitate a grid search….”

**Limitations:**

Yes, these have been discussed to the extend necessary.

---

> ### Author Rebuttal · Authors · 2024-08-05
>
> We thank the reviewer for the valuable feedback. We are glad to know that the reviewer appreciated our work from multiple aspects. We address the questions below, and we look forward to interacting with the reviewer during the discussion period.
>
> > “L77: it is easy to see..” Aren’t there cases where the optimization problem is still computationally, if not statistically, intractable?
>
> The reviewer is correct that even when the optimization problem is statistically tractable, it can still be computationally intractable. The recipe laid out in L77-79 relies on knowledge of the distribution map and an argmin oracle. Without the argmin oracle, it is often still possible to obtain a local minimum (with say first-order methods). We will clarify this in a revised version of the manuscript.
>
> > Could you provide a formal statement showing why one would still be able to tractably solve this modified version of the problem, assuming that the first version is tractable?
>
> To our understanding, the reviewer refers to PO as the first version of the problem, and DRPO as the modified version of the problem. As long as it is possible to at least solve the PO problem, then Algorithm 1 provides a way to reduce solving the DRPO problem to solving a sequence of PO problems.
>
> > Could you spell out in more detail exactly what the grid search is over when you state that L270 “for general problems, these calibration methods necessitate a grid search….”
>
> For the second calibration method "calibration set", the grid search performs the three steps listed between L265 and L268. For the first calibration method "post-fitting calibration", the grid search executes the following procedure according to the equations between L258 and L259: for a sorted candidate set of $\rho$'s, $\rho_1 < \rho_2 < \ldots < \rho_m$, we try $\rho_i~(1\leq i \leq m)$ from small to large and return the first $\rho_i$ such that $\max\_{\mathcal{D}\_{\operatorname{true}} \in \Xi} \widehat{D}(\mathcal{D}\_{\operatorname{true}}(\theta\_{\operatorname{DRPO}}(\rho\_i)) \|\| \mathcal{D}(\theta\_{\operatorname{DRPO}}(\rho\_i))) \leq \rho\_i$.

---

> > ### Comment · Reviewer_67u7 · 2024-08-07
> >
> > Thank you for the clarifications, stating out this reduction more clearly in the paper would be a good addition.

---

> > > ### Author Response · Authors · 2024-08-14
> > >
> > > Thank you for your kind suggestion:) We will properly incorporate this reduction viewpoint into Section 4.1.

---

### Official Review · Reviewer_UHwo · 2024-07-13

**Soundness:** 3
**Presentation:** 3
**Contribution:** 3
**Rating:** 6
**Confidence:** 4

**Summary:**

The paper addresses the performative prediction problem wherein the deployment of a model leads to a shift in the true data distribution via a distribution-shift map. In standard performative prediction, the distribution map is unknown - it is either assumed to be of a simple form like a location-scale shifting map, or estimated via a few deployments of models to see how the data is reacting. This paper proposes a distributionally robust performative prediction setting, where the objective is to minimize the distributionally robust performative risk (DRPR) which is the worst-cases performative risk over a set of distribution maps that contains the true distribution map. The uncertainty set is defined using KL divergence, and standard results from distributionally robust optimization (DRO) are used to obtain a dual form for DRPR. The paper then proposes to optimize the dual form of DRPR using an alternative minimization approach. The paper also proposes a simple calibration procedure to set the radius of uncertainty appropriately. The paper also proposes "tilted" performative risk minimization akin to the recently proposed tilted ERM (TERM). The proposed approach is validated via simulation on simple settings such as strategic classification with misspecified cost function, perfirmative prediction where the true distribution map is a linear location shifting functional.

**Strengths:**

- The framework of distributionally robust performative prediction is sound, and deserves attention.
- The presentation is clear for the most part.
- The regularizing effect of DRPR is, although not surprising, good to see.
- The connection made with TERM, and the proposed tilted performative risk minimization is interesting, and deserves further exploration.
- The experiments make a good case for the usefulness of the proposed approach.

**Weaknesses:**

- The theoretical results follow straightforwardly from existing results on distributionally robust optimization (DRO). The novelty is simply in bringing together the DRO framework and performative prediction.
- One of the steps in the proposed alternative minimization algorithm for DRPR itself is a performative risk minimization problem, which needs multiple deployments to solve. This means that the proposed DRPR framework needs even more deployments than standard performative risk minimization - it seems unrealistic for such an approach to work in a "real-world" setting.

**Questions:**

- Can you try to extend the result on regularizing effect of DRPR beyond the toy example in section 2? It seems like this should be possible.
- What is the effect of setting the parameter in tilted performative risk minimization to a negative value? It will be nice to compare this setting to the corresponding setting in TERM.

**Limitations:**

Yes, the authors have adequately addressed the limitations.

---

> ### Author Rebuttal · Authors · 2024-08-05
>
> We thank the reviewer for the valuable feedback. We are glad to know that the reviewer appreciated our work from multiple aspects. We address the questions and concerns below, and we look forward to interacting with the reviewer during the discussion period.
>
> > The novelty is simply in bringing together the DRO framework and performative prediction.
>
> The components of our approach are not new, but we are combining them in a novel way to solve a relevant problem. The novelty has been discussed in Section 6 between L371 and L375.
>
> > One of the steps in the proposed alternative minimization algorithm for DRPR itself is a performative risk minimization problem, which needs multiple deployments to solve.
>
> It depends on how one models the distribution map. When the distribution map is properly modeled (see the pushforward in Appendix E between L540 and L552), DRPR minimization doesn't require more deployments than standard PR minimization. For the most general case of distribution map modeling, Algorithm 1 needs additional cost of deployments.
>
> > Can you try to extend the result on regularizing effect of DRPR beyond the toy example in section 2? It seems like this should be possible.
>
> We extend the uni-variate example in Section 2 to its multi-variate case in Appendix A between L487 and L494. Regarding the general form of the loss function and distribution map, it is not expected that there will be a concise and elegant closed form formula, similar to the one found in the toy example.
>
> > What is the effect of setting the parameter in tilted performative risk minimization to a negative value? It will be nice to compare this setting to the corresponding setting in TERM.
>
> A negative tilt parameter in TERM suppresses the hard examples (the samples with high loss values) by assigning them less weight. When interpreting the hard examples as outliers, TERM with a negative tilt parameter is outlier-robust because it ignores the outliers to some degree. Similarly, tilted performative risk minimization with a negative tilt parameter can be viewed as an outlier-robust performative risk minimization approach. Unlike TERM, tilted performative risk minimization considers performativity. As a result, tilted performative risk minimization with a negative tilt parameter can be used when the distribution map (a family of distributions) is contaminated by a portion of outliers, but not just a single distribution. A negative tilt parameter, on the other hand, breaks the connection between the distributionally robust and tilted performative predictions. Because distributionally robust performative risk minimization focuses on hard examples but not ignoring them. In theory, the duality results necessitate a tilt parameter that is non-negative.

---

> > ### Author Response · Authors · 2024-08-14
> >
> > Thank you for taking the time to review our work! We did not hear from you during the author-reviewer discussion period. Please feel free to talk to the other reviewers and the area chair during the reviewer-AC discussion period:)

---

### Official Review · Reviewer_Zy5f · 2024-07-16

**Soundness:** 2
**Presentation:** 2
**Contribution:** 2
**Rating:** 4
**Confidence:** 2

**Summary:**

The paper proposes a framework that applies distributionally robust optimization to optimizing performative to obtain the distributionally robust performative optimum (DRPO). Specifically, the framework optimizes the worst case performative risk over the uncertainty set of distribution maps. The paper presents theoretical guarantees that bound the excess risk the approach as well as the benchmark approach. The paper also proposes tractable optimization algorithms for solving the distributionally robust problem and presents numerical simulations demonstrating the benefits of the proposed method.

**Strengths:**

The paper outlines a clear approach for applying distributionally robust optimization to the performative prediction/performative risk optimization setting. The descriptions of how to implement the framework well written. They also demonstrate how to obtain bounds on the excess performative risk to show the distributionally robust approach can perform as well as existing performative optimum approaches. The examples that were followed up upon in the numerics provided motivating examples of when applying the distributionally robust approach would be sensible.

**Weaknesses:**

The paper presents the distributionally robust optimization approach in an idealized setting where the form and structure of the distribution map is well defined and only a few key parameters need to be estimated. It would be more helpful to see how to arrive at this idealized setting from raw data. Overall, the paper could include more details about performative prediction to help readers understand how their approach fits into the bigger picture.

The paper's numerics and theoretical guarantees also do not allow readers to get a clear idea on how and if the distributionally robust approach improves excess risk. The theoretical guarantees while useful do not seem to necessarily be tight, so the improvement on the rates of the bounds does guarantee better empirical performance. Additionally, the numerics do not seem to study the excess risk metric (at least in the main body) which makes it hard to verify tightness of the theoretical guarantees and understand the practical benefits of the distributionally robust approach.

**Questions:**

1. The paper proposes two algorithms for the distributionally robust framework. Which algorithm is used in each of numeric examples and are there benefits using one over the other? Or is one algorithm just a special case of the other?
2. Can more insight be shed on the differences between the two excess risk bounds (prop 3.2 and prop 3.3)? In what settings are the two bounds and subsequently the excess risk significantly different?

**Limitations:**

The authors adequately addressed limitations and potential negative societal impact of their work.

---

> ### Author Rebuttal · Authors · 2024-08-05
>
> We thank the reviewer for the valuable feedback. We address the questions and concerns below, and we look forward to interacting with the reviewer during the discussion period.
>
> > Connection between numerics and theoretical guarantees.
>
> The numerical experiments and theoretical findings support the benefits of using DRPO over PO from different aspects. The theoretical findings demonstrate DRPO’s potential to outperform PO in achieving lower performative risk at a single distribution map. The numerical experiments show DRPO’s capacity to attain a more favorable worst-case performative risk across a collection of distribution maps, in comparison to PO.
>
> > The paper proposes two algorithms for the distributionally robust framework. Which algorithm is used in each of numeric examples and are there benefits using one over the other? Or is one algorithm just a special case of the other?
>
> The first experiment uses Algorithm 1 to find DRPO, while the second and third experiments use Algorithm 2 to find TPO. While both DRPO and TPO certify a certain level of robustness to misspecification of distribution maps, each has its own benefits over the other. On the one hand, DRPO exactly accounts for the uncertainty set size, whereas TPO does not. On the other hand, TPO is computationally more efficient, whereas DRPO requires more run time. The two algorithms are not special cases of each other. Their relationship should be understood as that the tilted performative risk minimization implicitly solves the corresponding DR performative risk minimization problem, and there is an implicit correspondence between the solution path $\theta_{\operatorname{DRPO}}(\rho)$ and $\theta_{\operatorname{TPO}}(\alpha)$.
>
> > Tightness of the theoretical guarantees? Can more insight be shed on the differences between the two excess risk bounds (prop 3.2 and prop 3.3)? In what settings are the two bounds and subsequently the excess risk significantly different?
>
> The excess risk bound for DRPO (3.3) is sharp in the asymptotic regime $\rho \to 0$. This is a consequence of the sensitivity property of KL divergence-based DRO. The excess risk bound for PO (3.2) is sharp in its rate, but the constant is likely loose. The key insight from the two bounds is the excess risk for DRPO has a localization property (the leading term of the excess risk vanishes even for fixed $\rho$) which is not shared by the excess risk of PO. This suggests that the DRPO bound is sharper than the PO bound.

---

> > ### Author Response · Authors · 2024-08-14
> >
> > Thank you for taking the time to review our work! We did not hear from you during the author-reviewer discussion period. Please feel free to talk to the other reviewers and the area chair during the reviewer-AC discussion period:)

---

### Official Review · Reviewer_bCrg · 2024-07-19

**Soundness:** 4
**Presentation:** 2
**Contribution:** 3
**Rating:** 6
**Confidence:** 4

**Summary:**

Summary: The paper revisits Performative Prediction, a setting proposed in 2020 where the chosen model used to minimize a prediction loss also induces a distribution shift in the data through a distribution map. When the true distribution map is unknown, the learner must use a nominal map to approximate the change of distribution resulting from the chosen model, but this may result in large errors. Using a distributionally robust formulation of the problem, they propose a way to guarantee that the learnt performative optimum has bounded error (if the chosen radius is well calibrated).

Overall, I believe the paper proposes good ideas and could be accepted. I have a few comments below, as well as some more minor ones about the writing. I will consider revising my score after the rebuttal.

**Strengths:**

Sound theoretical setting and analysis.

Classic but elegant algorithmic approach with practical extensions.

Empirical evaluation of all proposed methods.

**Weaknesses:**

Lack of discussions of the trade-offs / pros and cons.

"Toy" experiments on simple models.

--


Major remarks:

* Overall, the paper is not too hard to follow despite a number of slightly awkward writing issues that I’ve listed further below. However I am missing a real discussion section that puts into perspective the chosen approach and the obtained result. I would like to see a conclusion with a take-home message. I think the problem of not knowing the true distribution map is important but the distributionally robust approach chosen here is only one way to resolve it that focuses on worst-case guarantees. I think this is best shown in the experiment section but there are not many comments on it. For now the “summary and discussion” is essentially a summary with some pointers to extra results in appendix. What are the pros and cons of such approach? How general is it when the family of distribution maps is more complex than a linear function? When should one use this approach?

* Strategic classification example: Example 2.1 is a bit ill-defined because neither u nor c depend on \theta. In Section 5, it is a bit clearer as it is instantiated on a simple example (perhaps overly simple, is that common in this literature?). In the end the Kaggle data you cite is only used to estimate a base distribution but then all data is simulated, right? Figure 1 (left) shows that for a large interval of values of \epsilon_{true}, the performative risk obtained with the nominal map is actual lower than the distributional risk. I feel like the comment on this result l. 300-302 could be a bit expanded and insist on why this is typical for DR approaches.



Minor remarks:

* Your introduction clearly went through a beautiful GPT pass: “Delving deeper into the formulations of performative prediction, the concept of a distribution map emerges as pivotal.” While I generally don’t mind GPT assistance, I think this sentence does not carry much meaning. I think your introduction fails to really convey the motivation for distributionally robust guarantees, and how they differ from other existing guarantees. The sentences are complex and convoluted (e.g. “Practically, the precise influence of a model on the data ecosystem is intricate and dynamic, making perfect specification an unattainable ideal.”). I’d recommend making a pass sentence by sentence and making sure that everything is said efficiently and simply.

* Related work: “[28] study the repeated distributionally robust optimization algorithm, which repeatedly minimizes the distributionally robust risk at the induced distribution.” -> repetition, use \citet

* “The radius ρ reflects the the magnitude of shifts”-> double the

* “The dual reformulation (3.1) will be served as the cornerstone of developing algorithms” -> will serve? also “cornerstone of developing”? not sure about this sentence either

* Section 5 did not go through the same scrutiny of spell checks / GPT: l 315: “indentified”, l.324: ”estiamte”

* Please correctly use \citet and \citep everywhere, cf https://medium.com/@vincefort/phd-lessons-part2-ce830329c86f for guidelines, copied below for your convenience:
“Once you have an entry in your .bib file for a paper that you want to cite, say the key is author2022paper, you have two major ways of citing it: within the text or as a parenthesis. You can set the within-text citation as “Following \citet{author2022paper}, we do …” and it will render as “Following Author et al. (2022), we do …”. The parenthetical citation is set as “… this has been shown in prior work \citep{author2022paper}” and renders as “… this has been shown in prior work (Author et al. 2022)”. Make sure to not confuse the two types, as it can be a bit confusing for the reader and impair the flow.”

**Questions:**

What are the pros and cons of such approach? How general is it when the family of distribution maps is more complex than a linear function? When should one use this approach?

**Limitations:**

Experiments are on semi-simulations and use very simple model.

There is little discussion of the challenges of this DR approach when distribution maps are more complex (non-linear), high-dimensional.

---

> ### Author Rebuttal · Authors · 2024-08-05
>
> We thank the reviewer for the valuable feedback. We are glad to know that the reviewer appreciated our work from multiple aspects. We address the questions and concerns below, and we look forward to interacting with the reviewer during the discussion period.
>
> > What are the pros and cons of such approach? How general is it when the family of distribution maps is more complex than a linear function? When should one use this approach?
>
> The cons of our approach is additional computational cost. Because the true distribution map is unknown, the learner must rely on a nominal distribution map estimated from data, resulting in modeling or estimation errors. Due to this reason, as long as the additional computational cost is affordable, one should always use our approach, as the pros of our approach is that it is robust to distribution map misspecification. The linear distribution map is a standard setting that people consider in the research area of performative prediction (see simulations in Section 5.2 of [Perdomo et al. (2020)](https://arxiv.org/pdf/2002.06673)). However, our approach continues to work for more complex distribution maps.
>
>
> > Experiments are on semi-simulations and use very simple model. Is that common in this literature?
>
> Yes, that is common in the literature. Our experiments align with the established convention in performative prediction research area, which utilizes synthetic/semi-synthetic data and simple model with almost no exceptions. A semi-synthetic dataset is often a real-world data simulator, which comprises real data as the base distribution and a synthetic performativity setup. The usage of semi-synthetic data is common in the literature because it is difficult to come by a dataset to learn the real distribution map.
>
> > There is little discussion of the challenges of this DR approach when distribution maps are more complex (non-linear), high-dimensional.
>
> The theory and algorithms of our DR approach extend to the non-linear and high-dimensional cases. Conceptually, when dealing with non-linear and high-dimensional distribution maps, our approach can provide more benefits because distribution map estimation errors can be larger.
>
> > Example 2.1 is a bit ill-defined because neither u nor c depend on \theta. In Section 5, it is a bit clearer as it is instantiated on a simple example.
>
> We will change $u(x)$ to $u_{\theta}(x)$ to emphasize that the utility function depends on the model parameter. As the reviewer pointed out, this is actually instantiated in an example in Section 5 (particularly Subsection 5.1).
>
> > In the end the Kaggle data you cite is only used to estimate a base distribution but then all data is simulated, right?
>
> The reviewer is correct that the data is semi-synthetic: the base distribution comes from the Kaggle data, whereas the performativity setup is synthetic.
>
> > Figure 1 (left) shows that for a large interval of values of \epsilon_{true}, the performative risk obtained with the nominal map is actual lower than the distributional risk. I feel like the comment on this result l.
>
> Figure 1 (left) shows the curves for some discrete values of $\rho$'s. We didn't show the result of an "infinitesimal" $\rho$ (i.e. a $\rho$ extremely close to $0$), of which the performative risk curve should be "almost surely" slightly lower than that obtained with the nominal map, except for the point at $\epsilon_{\operatorname{true}} = 0.5$. Now let's consider a continuous spectrum of $\rho$'s. For values of $\rho$ ranging from small to moderate, DRPO outperforms PO in terms of performative risk (and similarly for worst-case performative risk). Conversely, for large values of $\rho$, PO is better than DRPO. There exists an “sweet spot” of $\rho$ where DRPO yields maximal benefits over PO. This trade-off between DRPO and PO is demonstrated in any “vertical slices” of the left plot of Figure 1 (and similarly in the lines of the middle plot of Figure 1 for worst-case performance).
>
> > 300-302 could be a bit expanded and insist on why this is typical for DR approaches.
>
> As $\rho$ increases, the DRPO aims to achieve low worst-case performative risk over a wider range of $\epsilon_{\operatorname{true}}$. This requires a trade-off between regions of $\epsilon_{\operatorname{true}}$ with low and high performative risk. Figure 1 (left) shows that as $\rho$ increases, the DRPO achieves more uniform performance ("flatter" performative risk curve) across a wider range of $\epsilon_{\operatorname{true}}$.

---

> > ### Comment · Reviewer_bCrg · 2024-08-12
> > **Thanks for your reply, please consider adding these discussions in the paper**
> >
> > Thank you for your replies, I remain in favour of accepting this paper. Please consider adding some of these clarifications to the paper, in particular the first questions (pros and cons, etc.). I just feel like robustness is an interesting addition to Performative prediction but it comes at a cost, it has limitations, that would be worth discussing.

---

> > > ### Author Response · Authors · 2024-08-14
> > >
> > > Thank you for your kind suggestion:) We will properly incorporate the discussion of the first set of questions (pros and cons, etc.) into Section 6, as well as other discussions into Appendix I.

---

### Decision · Program_Chairs · 2024-09-25

**Decision:**

Accept (poster)

**Comment:**

The paper revisits the performative learning setting in which the learner wishes to control the risk of a model over a test distribution that is sensitive to the choice of the model that is deployed. This is a novel niche topic but certainly an interesting and challenging setting that is connected to more well-known questions in ML such as covariate shift, lifelong ML or robust estimation. More precisely, the paper draws connections with distributionally robust optimization by studying methods to determine the worst-case performative risk over a set of distribution maps (that map parameters to test distributions) that contains the true distribution map.

While staying technically close both to ideas found in the original papers on performative prediction and to techniques originating from the distributionally robust optimization literature, this paper introduces novel ideas on a recent topic of interest which justify acceptance for the conference.